# Post-hoc Estimators for Learning to Defer to an Expert

**Harikrishna Narasimhan**
Google Research, Mountain View
hnarasimhan@google.com

**Wittawat Jitkrittum**
Google Research, New York
wittawat@google.com

**Aditya Krishna Menon**
Google Research, New York
adityakmenon@google.com

**Ankit Singh Rawat**
Google Research, New York
ankitsrawat@google.com

**Sanjiv Kumar**
Google Research, New York
sanjivk@google.com

## Abstract

Many practical settings allow a classifier to *defer* predictions to one or more costly *experts*. For example, the *learning to defer* paradigm allows a classifier to defer to a human expert, at some monetary cost. Similarly, the *adaptive inference* paradigm allows a base model to defer to one or more large models, at some computational cost. The goal in these settings is to learn classification and deferral mechanisms to optimise a suitable accuracy-cost tradeoff. To achieve this, a central issue studied in prior work is the design of a coherent loss function for both mechanisms. In this work, we demonstrate that existing losses can *underfit* the training set when there is a non-trivial deferral cost, owing to an implicit application of a high level of label smoothing. To resolve this, we propose two *post-hoc estimators* that fit a deferral function on top of a base model, either by threshold correction, or by learning when the base model's error rate exceeds the cost of deferring to the expert. Both approaches are equipped with theoretical guarantees, and empirically yield effective accuracy-cost tradeoffs on learning to defer and adaptive inference benchmarks.

## 1 Introduction

Supervised classification conventionally considers learning a *single* model with good average-case test performance. However, many practical settings allow for choosing amongst *multiple* models to classify a given sample. For example, in the *learning to defer* (*L2D*) paradigm [18], a *base* classifier may be used in conjunction with a human *expert*. Specifically, the classifier has the option of *deferring* to the expert, at the expense of incurring some additional cost (e.g., monetary expense). The goal is to learn a classifier and deferral function achieving a suitable accuracy-cost tradeoff. This problem has received a surge of interest [26, 19, 22, 34] given the increasing use of machine learning in high-stakes decision making. Interestingly, similar considerations are also found in the problem of *adaptive inference* for resource-constrained prediction [13]: here, a base model can choose to defer to one or more experts, which are themselves learning models with higher capacity and inference cost. Similar to L2D, the goal is to achieve a suitable tradeoff between accuracy and (inference) cost.

A key problem in such settings is deciding when the base model should defer its prediction to the expert(s). An intuitive strategy is to defer when the model has low *prediction confidence* (e.g., classification margin). While theoretically grounded [8], this strategy can under-perform when the base model has low capacity [9]. An alternate strategy is to explicitly formulate and optimise a joint loss function for the base classifier and deferral function. This has been followed with success in the L2D literature, particularly with the cost-sensitive softmax cross-entropy loss (`CSS`) of Mozannar and Sontag [19], and one-versus-all (`OvA`) loss of Verma and Nalisnick [34].

36th Conference on Neural Information Processing Systems (NeurIPS 2022).

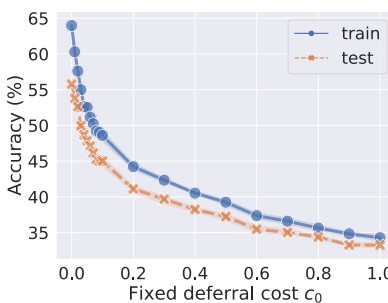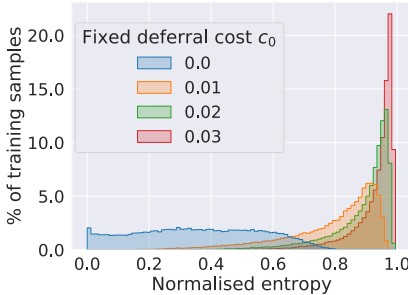

Figure 1: Illustration of underfitting of the cost-sensitive softmax cross-entropy (CSS) loss of Mozannar and Sontag [19]. On CIFAR-100, we consider a learning to defer setting comprising a ResNet-8 base model and a ResNet-32 "expert" $h_{\exp}$. We assume a cost $c_{\exp}(x, y) = c_0 + \mathbf{1}(y \neq h_{\exp}(x))$ of deferring to the expert (see §2 for details on notation), where the *fixed cost* $c_0$ is varied from $[0, 1]$. For each $c_0$, we train the base model using the CSS loss (4), and report the resulting accuracy. For $c_0 > 0$, the base model exhibits *underfitting*, evidenced by significant degradation in the *training* accuracy. In §3.1, we trace this behaviour to the loss applying a high level of *label smoothing* [31] to incorrect labels. Consequently, the *entropy* of the base model probabilities steadily increase with $c_0$ (right panel).

While the above losses have proven effective, they are not without limitations. Indeed, both focus on the setting where the cost of deferring is only given by the probability that the expert makes a mistake on a sample; this does not consider the additional *fixed cost* of querying an expert in the first place. As we shall see (§3), incorporating any non-zero fixed cost makes these losses prone to *underfitting*; cf. Figure 1. This behaviour is owing to the fact that the losses implicitly apply a high level of *label smoothing*. The setting of non-zero fixed cost is particularly important in adaptive inference settings, where querying an expert inherently involves paying an increased computational cost.

In this work, we resolve this issue via deferral schemes based on *post-hoc estimators*. In a nutshell, we follow a two-step procedure wherein we fit a deferral function on top of a base model, either by threshold correction, or by learning to predict when the base model's error rate exceeds the deferral cost. This allows us to build on top of the standard (zero fixed cost) CSS and OvA solutions, resulting in an empirically effective strategy with theoretical guarantees. In sum, our contributions are:

(i) we identify limitations in prior losses for L2D, particularly when the cost of deferring to an expert involves a non-zero fixed cost in addition to the probability of the expert making a mistake (§3)

(ii) we propose new post-hoc estimators for learning a deferral function, which overcome the above limitation while enjoying theoretical guarantees (§4, Theorem 2, Theorem 4)

(iii) we empirically show that the new post-hoc estimators work well on synthetic and real-world benchmark datasets for both L2D and adaptive inference settings (§5).

In the course of our analysis, we additionally generalise the OvA loss to reliably account for general deferral costs ((10), Lemma 1), and to reliably estimate label probabilities in practical settings (12).

## 2 Background and Notation

Our interest is in learning settings where a classifier is equipped with the option of deferring to an expert model. For concreteness, we shall focus our exposition on the learning to defer paradigm.

### 2.1 Multi-class Classification

Fix an instance space $\mathcal{X}$ and label space $\mathcal{Y} = [L] \doteq \{1, 2, \ldots, L\}$, and let $\mathbb{P}$ be a distribution over $\mathcal{X} \times \mathcal{Y}$. Given a training sample $S = \{(x_n, y_n)\}_{n \in [N]}$ drawn from $\mathbb{P}$, multi-class classification seeks a *classifier* $h \colon \mathcal{X} \to \mathcal{Y}$ with low *misclassification error* $R(h) = \mathbb{P}(y \neq h(x))$. We may parameterise $h$ as $h(x) = \operatorname{argmax}_{y' \in \mathcal{Y}} f_{y'}(x)$ for a *scorer* $f \colon \mathcal{X} \to \mathbb{R}^L$. Such a scorer may be learned by minimising the *empirical surrogate risk* $\hat{R}(f) = \frac{1}{N} \sum_{n \in [N]} \ell(x_n, y_n, f(x_n))$ for some *loss* $\ell \colon \mathcal{X} \times \mathcal{Y} \times \mathbb{R}^L \to \mathbb{R}_+$, such as the *softmax cross-entropy* $\ell(x, y, f(x)) = \log \left[ \sum_{y' \in \mathcal{Y}} e^{f_{y'}(x)} \right] - f_y(x)$. For binary classification problems with $L = 2$, it is common to enforce $f_2(x) = -f_1(x)$ and set $\ell(x, y, f(x)) = \phi(f_y(x))$ for some *margin loss* $\phi \colon \mathbb{R} \to \mathbb{R}_+$, such as $\phi(z) = \log(1 + e^{-z})$.

## 2.2 Learning to Defer (L2D) Problem

In the *learning to defer* (*L2D*) problem [18], one seeks a classifier that can either make a standard prediction in $\mathcal{Y}$ via some *base* model, or *defer* its prediction to an *expert* model $h_{\exp} \colon \mathcal{X} \to \mathcal{Y}$, at the expense of incurring a sample-dependent *cost* $c_{\exp}(x, y) > 0$. Typically, $c_{\exp}$ takes into account both the *fixed cost* of deferring to the expert, which we will assume to be a constant $c_0 \in [0, 1]$ (e.g., normalised monetary cost), and an estimate of the probability that the expert makes a mistake on the example. Canonical examples include $c_{\exp}^{(1)}(x, y) = c_0$ and $c_{\exp}^{(2)}(x, y) = c_0 + \mathbf{1}\,(y \neq h_{\exp}(x))$. The choice $c_{\exp}^{(1)}(x, y)$ corresponds to learning with a *reject option* or *abstention* [8, 2, 9, 6].

Given a suitable $c_{\exp}(x, y)$, the learning problem may be formalised as follows. Consider a classifier $\bar{h} \colon \mathcal{X} \to \mathcal{Y} \cup \{\perp\}$ equipped with a "defer" option $\perp$. Our goal is to minimise

$$R_{\mathrm{def}}(\bar{h}) = \mathbb{E}\left[\mathbf{1}(y \neq \bar{h}(x)) \cdot \mathbf{1}(\bar{h}(x) \neq \perp) + c_{\exp}(x, y) \cdot \mathbf{1}(\bar{h}(x) = \perp)\right]. \tag{1}$$

Intuitively, when $\bar{h}$ chooses not to defer, we incur the usual misclassification error; otherwise, we incur the cost $c_{\exp}(x, y)$ of invoking the expert $h_{\exp}$. One can cast (1) as an instantiation of *cost-sensitive learning* [11]; see Appendix B. We remark that Okati et al. [22] considered a slightly different setting, wherein one enforces a *hard* constraint on the expert cost $c_{\exp}(x, y)$. We focus on a *soft* additive penalty in (1), but we may handle hard constraints via standard Lagrangian theory; see Appendix G.

In practice, it is useful to parameterise $\bar{h}$ in terms of a scorer $\bar{f} \colon \mathcal{X} \to \mathbb{R}^{L+1}$ with label logits $\{\bar{f}_1(x), \ldots, \bar{f}_L(x)\}$, and "defer" logit $\bar{f}_\perp(x)$. We then predict via

$$\bar{h}(x) = \operatorname{argmax}_{y \in \mathcal{Y}} \bar{f}_y(x) \text{ if } \max_{y \in \mathcal{Y}} \bar{f}_y(x) > \bar{f}_\perp(x); \quad \bar{h}(x) = \perp \text{ otherwise.} \tag{2}$$

In practice, we may parameterise $\bar{f}$ via neural network logits: for any $y' \in \mathcal{Y} \cup \{\perp\}$, $\bar{f}_{y'}(x) = w_{y'}^\top \Phi(x)$ for weights $w_{y'}$ and embedding $\Phi$. This parameterisation induces implicit sharing amongst all logits.

One conceptually useful quantity is the *Bayes-optimal* $\bar{h}^* = \operatorname{argmin}_{\bar{h} \colon \mathcal{X} \to \mathcal{Y} \cup \{\perp\}} R_{\mathrm{def}}(\bar{h})$, which is

$$\bar{h}^*(x) = \begin{cases} \operatorname*{argmax}_{y \in \mathcal{Y}} \mathbb{P}(y \mid x) & \text{if } 1 - \max_{y \in \mathcal{Y}} \mathbb{P}(y \mid x) < \mathbb{E}_{y|x}\left[c_{\exp}(x, y)\right] \\ \perp & \text{else.} \end{cases} \tag{3}$$

Intuitively, we defer iff the expected cost of querying the expert is smaller than the expected error of the base model. For the constant cost function $c_{\exp}^{(1)}(x, y)$, this is known as *Chow's rule* [8].

## 2.3 Losses for Learning to Defer

We review some popular recent strategies to learn $\bar{f}$ (and thus $\bar{h}$) below.

**Confidence thresholding**. For $c_{\exp}(x, y) = c_0$, one strategy is to learn $\bar{f}_1, \ldots, \bar{f}_L$ by minimising the softmax cross-entropy, and fix $\bar{f}_\perp(x) = 1 - c_0$. This *confidence thresholding* approach [21] mimics the Bayes-optimal classifier (3). A related series of approaches [27, 38, 1] make deferral decisions by comparing the base model confidence with the confidence in the expert being correct. When the base model has low-capacity, such approaches may be sub-optimal as the base model may sub-optimally allocate its finite capacity by classifying examples that the expert can easily predict.

**Cost-sensitive softmax cross-entropy** (CSS). To overcome the limitations of confidence thresholding, it is desirable to explicitly learn $\bar{f}_\perp$ as well. To this end, Mozannar and Sontag [19] proposed a *cost-sensitive softmax cross-entropy* (CSS) loss: for a constant upper bound $c_{\max} \geq 1$ on the classification costs, this can be written as:

$$\ell_{\mathrm{CSS}}(x, y, \bar{f}(x)) = -\sum_{y' \in \mathcal{Y}} (c_{\max} - \mathbf{1}(y \neq y')) \cdot \log\left(\bar{p}_{y'}(x)\right) - (c_{\max} - c_{\exp}(x, y)) \cdot \log\left(\bar{p}_\perp(x)\right), \tag{4}$$

where $\bar{p}_y(x) = \exp(\bar{f}_y(x))/\sum_{y' \in \mathcal{Y} \cup \{\perp\}} \exp(\bar{f}_{y'}(x))$ is the softmax distribution over the standard *and* the defer label. For $c_{\exp}(x, y) = c_0 + \mathbf{1}\,(y \neq h_{\exp}(x))$, we have $c_{\max} = 1 + c_0$. The loss $\ell_{\mathrm{CSS}}$ is Fisher-consistent [19, Proposition 1], with Bayes-optimal prediction

$$(\forall y' \in \mathcal{Y})\ \bar{p}_{y'}^*(x) \propto c_{\max} - (1 - \mathbb{P}(y' \mid x)) \qquad \bar{p}_\perp^*(x) \propto c_{\max} - \mathbb{E}_{y|x}[c_{\exp}(x, y)]. \tag{5}$$

The original loss of Mozannar and Sontag [19] uses a slightly tighter formulation, with an upper bound $c_{\max}(x, y) \geq 1$ that is sample-dependent. Consequently, we can set $c_{\max}(x, y) = \max\{1, c_{\exp}(x, y)\}$. For simplicity, we use a constant upper bound for all the losses in this paper.

**One-versus-all (OvA).** Despite the Fisher consistency of $\ell_{\text{CSS}}$, Verma and Nalisnick [34] observed that empirically, the loss may not produce *calibrated* estimates of the expert's error probability. Verma and Nalisnick thus proposed a *one-versus-all* (OvA) loss for the cost function $c_{\exp}(x, y) = \mathbf{1}(y \neq h_{\exp}(x))$:

$$\ell_{\text{OVA}}(x, y, \bar{f}(x)) = \phi(\bar{f}_y(x)) + \sum_{y' \in \mathcal{Y} - \{y\}} \phi(-\bar{f}_{y'}(x)) + \tag{6}$$
$$(1 - c_{\exp}(x, y)) \cdot \phi(\bar{f}_\perp(x)) + c_{\exp}(x, y) \cdot \phi(-\bar{f}_\perp(x)).$$

Here, $\phi : \mathbb{R} \to \mathbb{R}_+$ is a binary *proper composite* loss [28], e.g., $\phi(z) = \log(1 + e^{-z})$. Such losses have an associated *inverse link function* $\psi \colon \mathbb{R} \to [0, 1]$, and produce Bayes-optimal scorer

$$(\forall y' \in \mathcal{Y}) \, \psi(\bar{f}_{y'}^*(x)) = \mathbb{P}(y' \mid x) \qquad \psi(\bar{f}_\perp^*(x)) = 1 - \mathbb{E}_{y|x}[c_{\exp}(x, y)]. \tag{7}$$

Intuitively, both (4) and (6) seek to have the first $L$ logits $\{\bar{f}_1(x), \ldots, \bar{f}_L(x)\}$ estimate $\mathbb{P}(y \mid x)$, and the reject logit $\bar{f}_\perp(x)$ model the cost of querying the expert. With this, the highest scoring logit provides us with an estimate of the Bayes-optimal deferral decision (3).

# 3 Limitations of Existing Learning to Defer Losses

We now study existing losses for learning to defer more carefully, and show that they may *underfit* in an important practical setting. Specifically, recall that a canonical choice of deferral cost function in (1) is $c_{\exp}(x, y) = \mathbf{1}(y \neq h_{\exp}(x)) + c_0$, for constant $c_0 \geq 0$. Here, $c_0$ represents a *fixed cost* of querying the expert, which is independent of whether the expert misclassifies the sample.

The setting of *non-zero fixed cost* $c_0 > 0$ is of practical interest; e.g., this is intrinsic to adaptive inference settings, wherein querying the expert model fundamentally involves increasing a computational overhead (see §4.3 for more discussion). Thus, an ideal surrogate loss for (1) should be performant for $c_0 > 0$. Unfortunately, we now show that this does *not* hold for the CSS and OvA losses.

## 3.1 Underfitting of Cost-Sensitive Softmax Cross-Entropy when $c_0 > 0$

The cost-sensitive softmax cross-entropy (CSS) is in principle applicable for any deferral cost $c_{\exp}(x, y)$, and in particular $c_{\exp}(x, y) = c_0 + \mathbf{1}(y \neq h_{\exp}(x))$ for any $c_0 \geq 0$. However, the focus in Mozannar and Sontag [19] was the case $c_0 = 0$. In fact, we now show that when $c_0 > 0$, the loss can *underfit*. To see this, note that the tightest possible maximal cost is $c_{\max} = 1 + c_0$. With this choice, (4) becomes

$$\ell_{\text{CSS}}(x, y, \bar{f}(x)) = -\log(\bar{p}_y(x)) + c_0 \cdot \sum_{y'} -\log(\bar{p}_{y'}(x)) - \mathbf{1}(y = h_{\exp}(x)) \cdot \log(\bar{p}_\perp(x)), \tag{8}$$

When $c_0 = 0$, the second term vanishes, and we have a conic combination of the log-loss on the true and deferral labels. When $c_0 > 0$, the second term is active, and applies a form of *label smoothing* [31]: we allow for *all* labels $y' \in \mathcal{Y} - \{y\}$ to be potential "positives" for $x$, with a weight of $c_0 \in [0, 1]$. Put differently, we treat the ground-truth label *distribution* as a mixture of a one-hot distribution on $y$, and a uniform distribution over all labels. As $c_0 \to 1^-$, we apply a large amount of smoothing on labels other than $y$; this can be problematic. Concretely, consider an $x$ where the expert gives the wrong prediction, i.e., $\mathbb{P}(y = h_{\exp}(x)) = 0$. Then, the Bayes-optimal solution (5) will be

$$(\forall y' \in \mathcal{Y}) \, \bar{p}_{y'}^*(x) = (\mathbb{P}(y' \mid x) + c_0)/(1 + L \cdot c_0) \qquad \bar{p}_\perp^*(x) = 0. \tag{9}$$

Observe that for $c_0 > 0$, as $L$ increases, the range of values in $\bar{p}^*$ shrinks: indeed, the gap between the optimal probability for true label $y$ and a competing label $y'$ will be $\mathcal{O}(1/L)$. This small *probability margin* makes learning challenging: given a finite sample, it will be easy for the model to mistakenly swap the true label $y$ with some competing label $y'$. Thus, in settings where $L$ is moderately large, and where there is a non-zero fixed deferral cost $c_0 > 0$, $\ell_{\text{CSS}}$ may underperform. Such issues have also been observed [7] for a similar loss developed for label noise [25].

As an illustration, we consider in Figure 1 a learning to defer setting comprising a ResNet-8 base model and a ResNet-32 "expert" on CIFAR-100. We assume a deferral cost $c_{\exp}(x, y) = c_0 + \mathbf{1}(y \neq h_{\exp}(x))$,

where $c_0$ is varied in $[0, 1]$. For each $c_0$, we train the base model using (4), and report the train and test accuracy. As $c_0$ increases, the base model shows significant *training* accuracy degradation, i.e., it underfits. The high level of label smoothing plays a role: in the right panel, we see that the *entropy* of the model's predicted distribution over labels increases with $c_0$. (The entropy is normalised by $\log L$ to lie in $[0, 1]$.) See Appendix I for additional plots, and Appendix I.3 for results when $c_{\exp}(x, y) = c_0$.

It is worth pointing out that the original loss of Mozannar and Sontag [19], which uses a tighter sample-dependent upper bound, will not have the label smoothing term on the samples that the expert predicts correctly, but will still underfit on the samples that the expert is wrong on.

## 3.2 Underfitting of One-Versus-All Loss when $c_0 > 0$

The one-versus-all (OvA) loss of Verma and Nalisnick [34] was presented for the specific case of $c_0 = 0$; as stated, the loss is not applicable when $c_0 > 0$. We may however extend this loss to a general $c_{\exp}(x, y)$ with upper bound $c_{\max} \geq 1$ (see Appendix D for a more general derivation):

$$\ell_{\text{OVA}}(x, y, \bar{f}(x)) = c_{\max} \cdot \phi(\bar{f}_y(x)) + \sum_{y' \neq y} \left( (c_{\max} - 1) \cdot \phi(\bar{f}_{y'}(x)) + \phi(-\bar{f}_{y'}(x)) \right)$$
$$+ (c_{\max} - c_{\exp}(x, y)) \cdot \phi(\bar{f}_{\perp}(x)) + c_{\exp}(x, y) \cdot \phi(-\bar{f}_{\perp}(x)). \quad (10)$$

Compared to (6), we have an additional weighted sum over each $\phi_{y'}(x)$. As with OvA, this loss results in a coherent Bayes-optimal solution when $\phi$ is a strictly proper composite loss [5, 28]. Recall that such losses are characterised by an inverse link function $\psi \colon \mathbb{R} \to [0, 1]$, such as the logistic loss $\phi(z) = \log(1 + e^{-z})$ with sigmoid inverse link $\psi(v) = (1 + \exp(-v))^{-1}$.

**Lemma 1.** *Suppose $\phi$ is a strictly proper composite loss with inverse link function $\psi$. Then, the one-versus-all loss in* (10) *is calibrated for the learning to defer risk* (1), *with Bayes-optimal scorer*

$$(\forall y' \in \mathcal{Y}) \, \psi(\bar{f}_{y'}^*(x)) = \frac{c_{\max} - (1 - \mathbb{P}(y' \mid x))}{c_{\max}} \quad \psi(\bar{f}_{\perp}^*(x)) = \frac{c_{\max} - \mathbb{E}_{y|x} \left[ c_{exp}(x, y) \right]}{c_{\max}}.$$

Unlike the optimal scorer for the CSS loss, here $\sum_{y' \in [L]} \psi(\bar{f}_{y'}^*(x)) \neq 1$. Nonetheless, the highest scoring logit agrees with the Bayes-optimal classifier (3) for learning to defer.

Reviewing (10), we see that when $c_{\max} > 1$, we apply a form of label smoothing by additional terms of the form $\sum_{y' \neq y} \phi(\bar{f}_{y'}(x))$, i.e., we effectively treat all $y' \neq y$ as potential positives. As with the cost-sensitive softmax entropy, this can result in underfitting; recall that with $c_{\exp}(x, y) = c_0 + \mathbf{1}(y \neq h_{\exp}(x))$, we have an upper bound $c_{\max} = 1 + c_0 > 1$. We empirically confirm in Appendix I.2 that we indeed observe analogous underfitting behaviour as per the CSS loss in Figure 1.

The above shows that the CSS and OvA losses have a subtle limitation when $c_0 > 0$. This is not in conflict with the Fisher-consistency of these losses: the latter establishes that the asymptotic minimiser of these losses agrees with the Bayes-optimal classifier (3). Our analysis explicates that the precise *form* of these Bayes-optimal solutions may not be amenable to learning from a *finite sample*.

Since both losses seamlessly handle $c_0 = 0$, a natural question is whether one can leverage this solution to help guide the one for $c_0 > 0$. We now present post-approaches that do precisely this.

# 4 Post-hoc Approaches to Learning to Defer

We now present two simple *post-hoc* strategies for learning to defer, which build on the CSS and OvA solutions for $c_0 = 0$ via *threshold correction* and *re-training* the deferral function.

## 4.1 Post-hoc Threshold Correction

Our first post-hoc strategy (Figure 2) exploits the form of the Bayes-optimal classifier in (3): to mimic this classifier for the cost function $c_{\exp}(x, y) = \mathbf{1}(y \neq h_{\exp}(x)) + c_0$, it suffices to obtain estimates of $\mathbb{P}(y \mid x)$ and the expected deferral cost $\mathbb{E}[c_{\exp}(x, y)] = \mathbb{P}(y \neq h_{\exp}(x)) + c_0$. Consequently, suppose we learn models $\pi \colon \mathcal{X} \to [0, 1]^L$ and $\epsilon \colon \mathcal{X} \to [0, 1]$ estimating $\mathbb{P}(y \mid x)$ and $\mathbb{P}(y \neq h_{\exp}(x))$ respectively. Then, for any $c_0 \geq 0$, we can approximate (3) via the classifier

$$\bar{h}(x) = \text{argmax}_{y \in \mathcal{Y}} \, \pi_y(x) \text{ if } 1 - \max_{y \in \mathcal{Y}} \pi_y(x) < \epsilon(x) + c_0; \quad \bar{h}(x) = \perp \text{ else.} \quad (11)$$

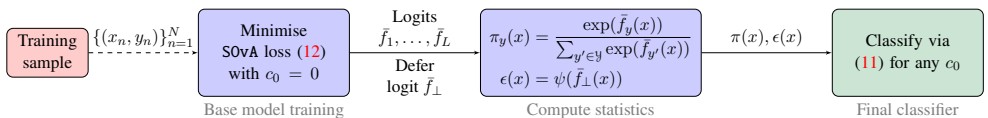

Figure 2: Summary of post-hoc threshold correction procedure (§4.1).

Intuitively, this procedure ought to perform well when $\pi(x)$ is close to $\eta(x) \doteq [\mathbb{P}(y \mid x)]_{y \in \mathcal{Y}}$, and $\epsilon(x)$ is close to $\mathbb{P}(y \neq h_{\exp}(x))$. The following *excess risk bound* for $\bar{h}$ formalises this intuition.

**Theorem 2.** *Pick any $\pi \colon \mathcal{X} \to [0,1]^L$ and $\epsilon \colon \mathcal{X} \to [0,1]$. Let $\bar{h}$ be the corresponding classifier in* (11). *Then, the excess risk of $\bar{h}$ compared to the Bayes-optimal classifier $\bar{h}^*$* (3) *is:*

$$R_{\text{def}}(\bar{h}) - R_{\text{def}}(\bar{h}^*) \leq c_{\max} \cdot \mathbb{E}_x \left[ \| \eta(x) - \pi(x) \|_1 \right] + 2 \cdot \mathbb{E}_x \left[ |\mathbb{P}(y \neq h_{\exp}(x)) - \epsilon(x)| \right].$$

Crucially, observe that we can learn $\pi$ and $\epsilon$ *once*, and then sweep across different $c_0$ values in (11). Thus, we can attempt to leverage the solutions for either the CSS or OvA losses with $c_0 = 0$ — recalling that these do *not* exhibit underfitting, unlike when $c_0 > 0$ — and then simply vary $c_0$ post-hoc. Indeed, both losses' Bayes-optimal solutions (5), (7) for $c_0 = 0$ suggest that a transform of $\bar{f}_1, \ldots, \bar{f}_L$ provide estimates of $\mathbb{P}(y \mid x)$, while a transform of $\bar{f}_\perp$ provides an estimate of $\mathbb{P}(y \neq h_{\exp}(x))$.

Unfortunately, in practice, both losses are potentially suboptimal to learn $\pi, \epsilon$. For the CSS loss, Verma and Nalisnick [34] showed it does not provide *calibrated* estimates of $\mathbb{P}(y \neq h_{\exp}(x))$. On the other hand, while the OvA loss provides calibrated estimates of $\mathbb{P}(y \neq h_{\exp}(x))$, its probability estimates of $\mathbb{P}(y \mid x)$ may underperform when $L$ is large (see Appendix J). We overcome this with a simple solution: we estimate $\pi$ via the softmax cross-entropy, and $\epsilon$ via the OvA loss. This is achieved via a *hybrid* softmax cross-entropy plus OvA loss (SOvA):

$$\ell_{\text{SOVA}}(x, y, \bar{f}(x)) = -\log p_y(x) + (c_{\max} - c_{\exp}(x, y)) \cdot \phi(\bar{f}_\perp(x)) + c_{\exp}(x, y) \cdot \phi(-\bar{f}_\perp(x)), \quad (12)$$

where $p_y(x) = \exp(\bar{f}_y(x)) / \sum_{y' \in \mathcal{Y}} \exp(\bar{f}_{y'}(x))$. Following the Bayes-optimal solution, we set $\pi(x) = (p_1(x), \ldots, p_L(x))$ and $\epsilon(x) = \psi(\bar{f}_\perp(x))$. As with the CSS and OvA losses, we may parameterise $\bar{f}_{y'}(x) = w_{y'}^\top \Phi(x)$ for shared $\Phi$, and thus induce information sharing amongst all logits.

The above post-hoc approach crucially relies on estimating the expert's error rate. A potentially simpler task is learning to predict whether it is *beneficial* to invoke the expert, i.e., whether the expert's error is lower than that of the base model. We now consider an alternate strategy that implements this.

## 4.2 Post-hoc Rejector Training

Our second post-hoc strategy revisits the fundamental risk (1) underpinning the learning to defer problem. While defined in terms of a classifier $\bar{h} \colon \mathcal{X} \to \mathcal{Y} \cup \{\perp\}$, we may rewrite this in terms of a *base classifier* $h \colon \mathcal{X} \to \mathcal{Y}$ and *rejector* $r \colon \mathcal{X} \to \mathbb{R}$, with the latter denoting the confidence in deferring:

$$R_{\text{def}}(h, r) = \mathbb{E}_{(x,y)}[\mathbf{1}(r(x) < 0) \cdot c_{\text{mod}}(x, y) + \mathbf{1}(r(x) > 0) \cdot c_{\exp}(x, y)], \quad (13)$$

where $c_{\text{mod}}(x, y) = \mathbf{1}(y \neq h(x))$. Given a scorer $\bar{f} \colon \mathcal{X} \to \mathbb{R}^{L+1}$, we may set $h(x) = \arg\max_{y' \in \mathcal{Y}} \bar{f}_{y'}(x)$ and $r(x) = \bar{f}_\perp(x) - \max_{y' \in \mathcal{Y}} \bar{f}_{y'}(x)$. Both CSS and OvA construct surrogate losses that are applied *jointly* over all logits $\bar{f}_1, \ldots, \bar{f}_L, \bar{f}_\perp$; equally, they construct a joint surrogate loss for both the base classifier $h$ and rejector $r$. As established earlier, the solution obtained from these losses is reasonable under fixed deferral cost $c_0 = 0$, but tends to suffer when $c_0 > 0$.

We may resolve this issue via a two-step procedure (see Figure 3): in the first step, we obtain standard logits $\bar{f}_1(x), \ldots, \bar{f}_L(x)$ by minimising either the CSS or OvA loss with $c_0 = 0$. In the second step, we train *only* the deferral logit, by constructing a *partial* surrogate for only the rejector $r(x)$ in (13). Concretely, for a binary surrogate loss $\phi \colon \mathbb{R} \to \mathbb{R}_+$, we may define the following surrogate for $r$:

$$\ell_{\text{rej}}(x, y, r(x); h) = c_{\text{mod}}(x, y) \cdot \phi(r(x)) + c_{\exp}(x, y) \cdot \phi(-r(x)), \quad (14)$$

and minimize the resulting partial surrogate risk over $r$:

$$R_{\text{def}}^\phi(h, r) = \mathbb{E}\left[\ell_{\text{rej}}(x, y, r(x); h)\right]. \quad (15)$$

With this loss, $c_0$ only plays a role in the $c_{\exp}(x, y)$ weighting term, and thus only influences the deferral decision; it does not induce any underfitting of the standard logits $\bar{f}_1, \ldots, \bar{f}_L$. Observe also

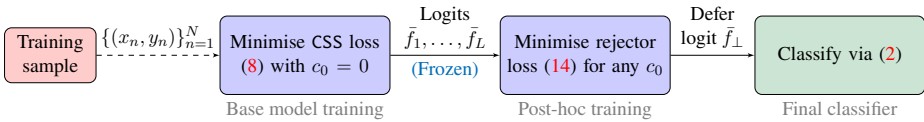

Figure 3: Summary of post-hoc training procedure (§4.2).

that $c_{\text{mod}}(x, y)$ will involve the *frozen* logits $\bar{f}_1, \ldots, \bar{f}_L$ obtained from the first stage. We may further leverage these logits by parameterising $r(x) = \bar{f}_\perp(x) - \max_{y' \in \mathcal{Y}} \bar{f}_{y'}(x)$, and optimising only the $\bar{f}_\perp(x)$ term. In practice, exploiting the *probabilities* from the first stage model can offer a closer approximation to the Bayes-optimal classifier; see Appendix F for more discussion.

Theoretically, the surrogate loss in (14) provides a calibrated deferral rule for any base classifier $h$. For strictly proper composite $\phi$ with inverse link $\psi \colon \mathbb{R} \to [0, 1]$, the optimal rejector is as follows.

**Lemma 3.** *The Bayes-optimal rejector for the loss* (14) *is* $\psi(r^*(x)) = \frac{\mathbb{E}_{y|x}[c_{mod}(x,y)]}{\mathbb{E}_{y|x}[c_{mod}(x,y)] + \mathbb{E}_{y|x}[c_{exp}(x,y)]}$.

Note that when the first stage logits agree with the Bayes-optimal solution, $\mathbb{E}_{y|x}[c_{\text{mod}}(x, y)] = 1 - \max_{y \in \mathcal{Y}} \mathbb{P}(y \mid x)$, and so deferring when $\psi(r^*(x)) > 0.5$ produces the optimal decision.

We next provide a bound an *excess risk bound* for a classifier learned using the post-hoc training procedure, under the assumption that the surrogate loss $\phi$ used is classification calibrated [3].

**Theorem 4.** *Suppose $\phi$ is classification calibrated and $c_{\exp}(x, y) \in [c_{\min}, c_{\max}]$ for some $c_{\min} > 0$. Denote the misclassification risk for classifier $h$ by $R_{\text{err}}(h) = \mathbb{E}[\mathbf{1}(y \neq h(x))]$. Let $\hat{r}$ be the rejector obtained by minimizing the partial surrogate risk in* (15) *for base classifier $\hat{h}$. Then the excess L2D risk for the resulting classifier $(\hat{h}, \hat{r})$ is bounded in terms of the excess surrogate risk for the rejector $\hat{r}$ and the excess misclassification risk for the base classifier $\hat{h}$:*

$$R_{\text{def}}(\hat{h}, \hat{r}) - \min_{\bar{h}: \mathcal{X} \to [L] \cup \{\perp\}} R_{\text{def}}(\bar{h})$$

$$\leq \Psi\left(R_{\text{def}}^\phi(\hat{h}, \hat{r}) - \min_{r: \mathcal{X} \to \mathbb{R}} R_{\text{def}}^\phi(\hat{h}, r)\right) + R_{\text{err}}(\hat{h}) - \min_{h: \mathcal{X} \to [L]} R_{\text{err}}(h),$$

*for some increasing function $\Psi : \mathbb{R}_+ \to \mathbb{R}_+$ with $\Psi(0) = 0$.*

### 4.3 Discussion and Extensions

**Relation to existing work**. Post-hoc threshold correction generalises Chow's rule [8] for learning to reject, wherein $c_{\exp}(x, y) = c_0$. It also generalises the approaches of Raghu et al. [27], Wilder et al. [37], Bansal et al. [1] for learning to defer: these assume $c_0 = 0$, and use *separate* models to estimate $\pi(x)$ and $\epsilon(x)$. By contrast, we estimate these quantities with a *single* neural model, by minimising the SOvA loss in (12) with $\bar{f}_{y'}(x) = w_{y'}^\top \Phi(x)$ for $y' \in \mathcal{Y} \cup \{\perp\}$. As with the standard OvA loss, this induces implicit information sharing between the defer logit $\bar{f}_\perp$ and the standard logits $\bar{f}_1, \ldots, \bar{f}_L$; thus, the estimates of the expert error probability will be influenced by the base model. Note that the confidence approach of Raghu et al. [27] was shown to be outperformed by the CSS and OvA losses in Mozannar and Sontag [19], Verma and Nalisnick [34]; our proposed threshold correction further achieves competitive or superior performance to these losses (§5).

Verma and Nalisnick [34, Appendix E] considered a post-hoc scheme for the setting where there is a hard constraint on the expert cost. Here, one trains a probabilistic model $\bar{p}$ using the OvA loss, and defers on the samples with the largest values of $\bar{p}_\perp(x) - \max_{y' \in \mathcal{Y}} \bar{p}_{y'}(x)$. While derived for a different setting, this is similar to (11), as varying the cost $c_0$ effectively changes the fraction of samples that are deferred. Note however that we use the SOvA rather than OvA loss to obtain $\pi, \epsilon$, which can have a notable impact on performance when $L$ is large (see Appendix J).

Cortes et al. [9] proposed a *joint* surrogate over $r, \bar{f}$ for (13) in the special case of constant $c_{\exp}(x, y) = c_0$, and a scalar scorer $\bar{f} \colon \mathcal{X} \to \mathbb{R}$ for binary labels. By contrast, the surrogate loss (14) handles multi-class $\bar{f}$, generic cost functions $c_{\exp}$, and provides a partial surrogate over only $r$.

**Adaptive inference and multiple experts**. Post-hoc training is naturally amenable to settings where there are *multiple* experts. Concretely, suppose there are $K$ experts with logits $\bar{f}^{(1)}(x), \ldots, \bar{f}^{(K)}(x)$,

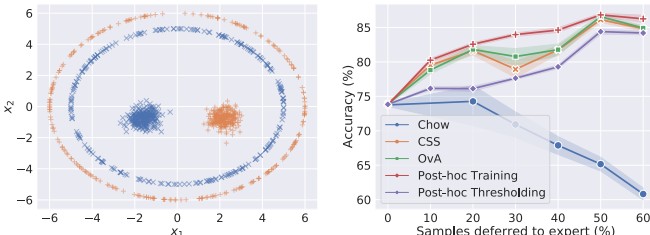

Figure 4: Results on synthetic dataset (left panel) comprising two subgroups, each with positive ($+$) and negative ($\times$) samples. The subgroup of concentric circles corresponds to a latent variable $a = 1$, while the subgroup of adjacent circles corresponds to $a = 0$. The base learner is a linear classifier on the raw features $x = (x_1, x_2)$, and can query an expert that trains a classifier on *quadratic* features. We train classifiers using different losses, for varying fixed deferral cost $c_0$. Posthoc training is competitive or superior to the cost-sensitive softmax (CSS) and one-versus-all loss (OvA) (right panel).

and our goal is to learn which expert to employ for a given sample. One may learn *multiple* deferral functions $\bar{f}_\perp^{(1)}(x), \ldots, \bar{f}_\perp^{(K)}(x)$ in a post-hoc manner, and choose the smallest $k$ with $\bar{f}_\perp^{(k)}(x) < 0$.

This setting arises naturally in *adaptive inference* [13], where our goal is to reduce the *inference cost* of a learning system by allowing for variable compute across samples: intuitively, we would like to spend less computation on "easy" samples. To achieve this, one may construct several models ("experts") of increasing inference cost, and defer to a suitable expert. Such models include *early-exit* classification heads at intermediate nodes of a network [24, 32, 36, 15, 29], or *sequential cascades* of models of increasing complexity [4, 30, 35]. Conceptually, adaptive inference involves the same core problem as learning to defer: at any given intermediate model, one has the option to defer prediction to the next model, at the expense of increasing the overall computation cost. Interestingly, similar ideas to post-hoc training have been explored in adaptive inference [33]. Compared to this work, we present a unified view of the learning to defer and adaptive inference problems, demonstrate limitations of existing losses for the former, and leverage generic logits $\bar{f}_1, \ldots, \bar{f}_L$ from any base classifier.

# 5 Experimental Results

We now present empirical results illustrating the efficacy of both our proposed posthoc estimators.

## 5.1 Results on Synthetic Data

We begin with a synthetic problem where $\mathcal{X} \subset \mathbb{R}^2$. We consider a distribution $\mathbb{P}(x, y) = \sum_{a \in \{0,1\}} \mathbb{P}(x, y, a)$, where $a \in \{0, 1\}$ denotes some latent subgroup. Figure 4 shows a sample draw from $\mathbb{P}(x, y)$: in a nutshell, $\mathbb{P}(x, y \mid a = 0)$ is a mixture of isotropic Gaussians, while $\mathbb{P}(x, y \mid a = 1)$ comprises two concentric circles. See Appendix H.1 for a precise specification. We consider learners that have access to a sample $\{(x_n, y_n)\}_{n=1}^N$ from $\mathbb{P}(x, y)$, and to expert model predictions $\{h_{\exp}(x_n)\}_{n=1}^N$. The expert is taken to be a linear classifier that is trained on only samples with $a = 1$, using *quadratic* features: i.e., for $x = (x_1, x_2)$, we construct $\Phi(x) = (x_1, x_2, x_1^2, x_2^2, x_1 \cdot x_2)$. We set the expert deferral cost function to be $c_{\exp}(x, y) = c_0 + \mathbf{1}(y \neq h_{\exp}(x))$.

We train linear classifiers using confidence thresholding based on a constant deferral cost $c_0$ (Chow), the CSS and OvA losses, and our proposed post-hoc approaches. For each learner, we vary the fixed deferral cost $c_0$ from $\{0.0, 0.05, 0.10, \ldots, 0.50\}$. For each setting, we compute the fraction of samples deferred to the expert, and the overall accuracy of the base model plus expert. We conduct 250 independent trials, each with a different draw of the training sample and expert predictions.

We highlight a few trends from the results in Figure 4. First, Chow strongly underperforms post-hoc thresholding when many samples are deferred. This is because the latter does not consider the expert's error (which is high on $a = 1$) when deferring. Second, post-hoc training is competitive or superior to CSS and OvA at most operating points. Third, post-hoc thresholding underperforms post-hoc training. This is by construction, as it is challenging to reliably estimate $\mathbb{P}(y \mid x)$ with a linear model. However, we shall now see that on real-world benchmarks, thresholding is far more competitive.

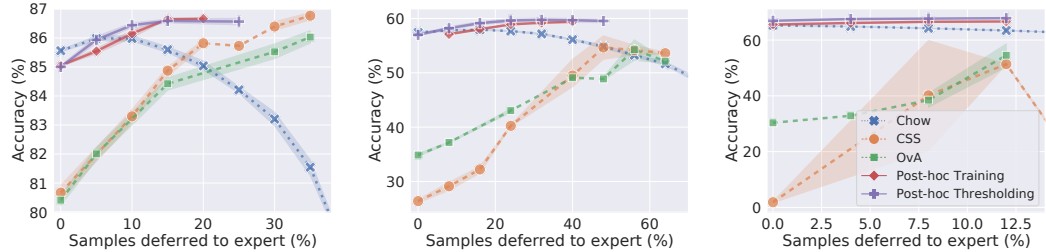

Figure 5: Results on CIFAR-10 (left), CIFAR-100 (middle), and ImageNet (right) in a learning to defer setting, where a base model is allowed to defer to a "specialist" expert. The latter is trained on only samples from the first 5 classes on CIFAR-10, the 50 classes from the first 10 "coarse labels" on CIFAR-100, and only classes corresponding to the "dog" synset on ImageNet. Our posthoc schemes offers gains over the existing cost-sensitive softmax (CSS) and one-versus-all (OvA) at most operating points, particularly when only a few samples are deferred to the expert (corresponding to fixed deferral cost $c_0 \gg 0$).

## 5.2 Results on Real-World Data

We now report results on the CIFAR-10, CIFAR-100 [16], and ImageNet [10] datasets.

**Learning to defer**. Inspired by Mozannar and Sontag [19, Section 6.2], we consider a setting where there is a "specialist" expert: we train an expert model on only samples with label belonging to some subset $\mathcal{Y}_{\text{sub}}$. Intuitively, this model "specialises" to only samples from $\mathcal{Y}_{\text{sub}}$. We then consider a base model that is trained on all samples, but is allowed to defer to the "specialist" expert. On CIFAR-10 and CIFAR-100, we use a ResNet-8 base model and ResNet-56 expert; on ImageNet, we use a MobileNet-v2 base model and an EfficientNet-B0 expert. The label subset $\mathcal{Y}_{\text{sub}}$ comprises the first 5 labels on CIFAR-10, the 50 labels corresponding to the first 10 "coarse labels" on CIFAR-100, and all labels corresponding to the "dog" synset on ImageNet.

We consider a cost $c_{\text{exp}}(x, y) = c_0 + \mathbf{1}(y \neq h_{\text{exp}}(x))$ of deferring to the expert model. Per Figures 2 and 3, we apply posthoc threshold correction on top of the SOvA solution with $c_0 = 0$, and posthoc training on top of the CSS solution with $c_0 = 0$. Figure 5 compares all approaches as we vary the fixed cost $c_0$, reporting the fraction of samples deferred to the expert, and the overall accuracy of the base model plus expert. Our posthoc estimators offer gains over the existing approaches at most operating points. We particularly see gains in the regime where only a small fraction of samples are deferred to the expert, which corresponds to larger values of $c_0$. Between the two post-hoc methods, we see that the threshold correction approach tends to have a (slight) edge. This suggests that on these benchmarks, it is possible to obtain sufficiently reliable estimates of $\mathbb{P}(y \mid x)$ and $\mathbb{P}(y \neq h_{\text{exp}}(x))$.

We reiterate that when $c_0$ is large, the existing CSS and OvA losses tend to degrade owing to underfitting. Indeed, their performance in this regime is significantly lower than simply using the student model trained with the softmax cross-entropy (corresponding to Chow with 0% of samples deferred). Note also that in comparison to the synthetic dataset, here $L$ is larger and thus further exacerbates underfitting. Finally, as in the synthetic dataset, Chow strongly underperforms when a large fraction of samples are deferred to the expert; this is because it does not take into account the expert's error rate, and thus potentially defers when the expert performs akin to random guessing.

**Adaptive inference**. We conclude with results on the related problem of adaptive inference. We consider a setting where there is a cascade of two models of differing inference costs, as measured in floating point operations (*FLOPs*), and we seek to learn a deferral function that intelligently forwards a small subset of samples to the latter model. This allows for an inference scheme with a favourable accuracy-compute tradeoff. Compared to the previous section, we now train the experts on *all* labels.

We report results on CIFAR-100 and ImageNet, with the former using ResNet-8 and ResNet-56 models, and the latter using a MobileNet-v2 and EfficientNet model. As before, we consider the deferral cost function $c_{\text{exp}}(x, y) = c_0 + \mathbf{1}(y \neq h_{\text{exp}}(x))$. Figure 6 compares the FLOPs versus the overall accuracy of the base model plus expert. Here, the CSS and OvA methods again degrade in the low FLOPs (high $c_0$) regime. However, the Chow baseline is more competitive, since the expert models tend to be highly accurate on all samples. Our posthoc estimators are seen to strongly outperform CSS and OvA, while being competitive with or slightly superior to Chow.

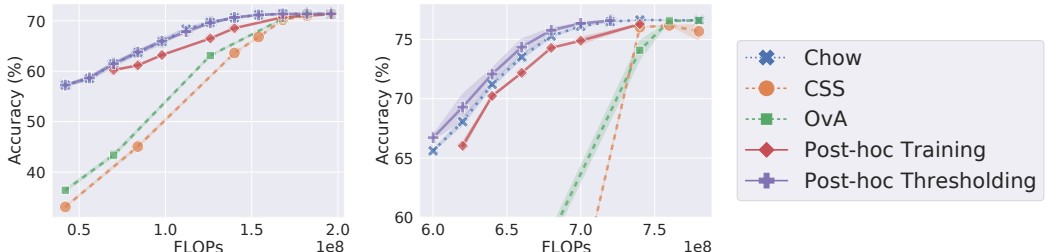

Figure 6: Results on CIFAR-100 (left), and ImageNet (right) in an adaptive inference setting, where a computationally cheap base model is allowed to defer to a more expensive expert. Our posthoc schemes offers gains over the existing cost-sensitive softmax (`CSS`) and one-versus-all (`OvA`) at most operating points, particularly in the low-FLOPs regime (corresponding to fixed deferral cost $c_0 \gg 0$).

## 6 Discussion and Future Work

While post-hoc training resolves issues identified with previous joint surrogate losses, it is not without limitation: it relies crucially on reliable base classifiers from the first phase. As formalised in Theorem 2, any degradation in performance for the latter is also transferred to the final post-hoc predictor. As this may be further compounded with estimation errors in the second phase, it is natural to ask whether there may be *other* joint surrogates that do not exhibit underfitting. One candidate is the cost-sensitive loss of Lee et al. [17], which is Fisher-consistent and is suitable when $c_0 > 0$. This loss imposes the constraint that $\sum_{\hat{y} \in \mathcal{Y} \cup \{\perp\}} \bar{f}_{\hat{y}}(x) = 0$, and takes the form

$\ell(x, y, \bar{f}(x)) = \sum_{y' \in \mathcal{Y} - \{y\}} \left[ \frac{1}{L-1} + \bar{f}_{y'}(x) \right]_+ + c_{\exp}(x, y) \cdot \left[ \frac{1}{L-1} + \bar{f}_{\perp}(x) \right]_+$. One may show that

under this loss, the gap between the Bayes-optial scores for the highest scoring label and any competing label is $\mathcal{O}\left(1 + \frac{1}{L}\right)$, and thus is not adversely affected as $L$ increases. As a qualifying comment, the loss poses challenges due to being non-differentiable, and requiring constraints on the logits. Nonetheless, further study of the viability of this loss would be worthwhile.

**Acknowledgment.** We thank Hussein Mozannar for pointing out that the original cost-sensitive softmax cross-entropy loss of Mozannar and Sontag [19] uses a slightly tighter formulation than the one we consider, and will not underfit on samples that the expert predicts correctly.

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
