# Supplementary material for "Post-hoc estimators for learning to defer to an expert"

## A  Proofs

### A.1  Proof of Lemma 1

*Proof.* To show that our proposed loss is calibrated for the cost function $c_{\exp}(x, y)$, we write out the conditional risk for the loss:

$$
\mathbb{E}_{y|x}\left[\ell(y, f(x))\right] = \sum_{y \in [L]} \eta_y(x) \left( c_{\max} \cdot \phi(f_y(x)) + \sum_{y' \neq y} \left((c_{\max} - 1) \cdot \phi(f_{y'}(x)) + \phi(-f_{y'}(x))\right) \right)
$$
$$
+ \left(c_{\max} - \mathbb{E}_{y|x}\left[c_{\exp}(x,y)\right]\right) \cdot \phi(f_\perp(x)) + \mathbb{E}_{y|x}\left[c_{\exp}(x,y)\right] \cdot \phi(-f_\perp(x))
$$
$$
= \sum_{y \in [L]} \eta_y(x) \left( \phi(f_y(x)) + \sum_{y' \in [L]} (c_{\max} - 1) \cdot \phi(f_{y'}(x)) + \sum_{y' \neq y} \phi(-f_{y'}(x)) \right)
$$
$$
+ \left(c_{\max} - \mathbb{E}_{y|x}\left[c_{\exp}(x,y)\right]\right) \cdot \phi(f_\perp(x)) + \mathbb{E}_{y|x}\left[c_{\exp}(x,y)\right] \cdot \phi(-f_\perp(x))
$$
$$
= \sum_{y \in [L]} \left(\eta_y(x) \cdot \phi(f_y(x)) + (1 - \eta_y(x)) \cdot \phi(-f_y(x))\right) + \sum_{y \in [L]} (c_{\max} - 1) \cdot \phi(f_y(x))
$$
$$
+ \left(c_{\max} - \mathbb{E}_{y|x}\left[c_{\exp}(x,y)\right]\right) \cdot \phi(f_\perp(x)) + \mathbb{E}_{y|x}\left[c_{\exp}(x,y)\right] \cdot \phi(-f_\perp(x))
$$
$$
= \sum_{y \in [L]} \left(\eta_y(x) + c_{\max} - 1\right) \cdot \phi(f_y(x)) + (1 - \eta_y(x)) \cdot \phi(-f_y(x))
$$
$$
+ \left(c_{\max} - \mathbb{E}_{y|x}\left[c_{\exp}(x,y)\right]\right) \cdot \phi(f_\perp(x)) + \mathbb{E}_{y|x}\left[c_{\exp}(x,y)\right] \cdot \phi(-f_\perp(x)).
$$

For any $y \in \mathcal{Y}$, the optimal scorer is thus

$$
f_y^*(x) = \operatorname*{argmin}_{v \in \mathbb{R}} a(x) \cdot \phi(v) + (1 - a(x)) \cdot \phi(-v)
$$
$$
a(x) \doteq \frac{\eta_y(x) + c_{\max} - 1}{c_{\max}}.
$$

Similarly, for the defer logit,

$$
f_\perp^*(x) = \operatorname*{argmin}_{v \in \mathbb{R}} b(x) \cdot \phi(v) + (1 - b(x)) \cdot \phi(-v)
$$
$$
b(x) \doteq \frac{c_{\max} - \mathbb{E}_{y|x}\left[c_{\exp}(x,y)\right]}{c_{\max}}.
$$

By definition of a strictly proper composite loss [28], the optimal scorer $f^*$ then takes the form:

$$
\psi(f_{y'}^*(x)) = \begin{cases} \frac{1}{c_{\max}}\left(\eta_{y'}(x) + c_{\max} - 1\right) & \text{if } y' \in [L] \\ \frac{1}{c_{\max}}\left(c_{\max} - \mathbb{E}_{y|x}\left[c(x,y)\right]\right) & \text{if } y' = \perp, \end{cases}
$$

where $\psi$ is the inverse link function associated with $\phi$, and the corresponding classifier $\bar{h}^*(x) \in \operatorname*{argmax}_{y' \in [L] \cup \{\perp\}} f_{y'}^*(x)$ is given by:

$$
\bar{h}^*(x) = \begin{cases} \operatorname*{argmax}_{y' \in [L]} \eta_{y'}(x) & \text{if } \max_{y' \in [L]} \eta_{y'}(x) < 1 - \mathbb{E}_{y|x}\left[c_{\exp}(x,y)\right] \\ \perp & \text{otherwise} \end{cases},
$$

as desired.

$\square$

### A.2  Proof of Theorem 2

*Proof.* The proof is similar to that of a regret bound in Narasimhan et al. [20, Lemma 14] for a general-cost-sensitive risk. As noted in §4.1, we denote $\eta_y(x) = \mathbb{P}(y \mid x)$ and $\eta(x) \doteq [\mathbb{P}(1 \mid x), \ldots, \mathbb{P}(L \mid$

$x)]^\top$. We will denote the expected cost of deferring to the expert using $C_{\exp}(x) \doteq \mathbb{E}_{y|x}[c_{\exp}(x,y)]$ and use $\hat{C}_{\exp}(x)$ to denote an estimate of this quantity.

We then re-write the learning to defer risk in (1) in terms of $\eta(x)$ and $C_{\exp}(x)$:

$$R_{\mathrm{def}}(h) = \mathbb{E}_x \left[ \sum_{y=1}^{L} \eta_y(x) \left( \mathbf{1}(h(x) \neq y) + C_{\exp}(x) \cdot \mathbf{1}(h(x) = \perp) \right) \right]$$

$$= \mathbb{E}_x \left[ \langle \eta(x), e_{h(x)}^{\mathrm{mod}} + e_{h(x)}^{\exp}(x) \rangle \right],$$

where $\langle \cdot, \cdot \rangle$ denotes standard inner-product, $e_{\hat{y}}^{\mathrm{mod}} = [\mathbf{1}(\hat{y} \neq 1), \ldots, \mathbf{1}(\hat{y} \neq L)]^\top \in \{0,1\}^L$ and $e_{\hat{y}}^{\exp}(x) = [C_{\exp}(x) \cdot \mathbf{1}(\hat{y} \neq \perp), \ldots, C_{\exp}(x) \cdot \mathbf{1}(\hat{y} \neq \perp)]^\top \in \mathbb{R}_+^L$.

We also define an empirical version of the learning to defer risk, in which $\eta(x)$ is replaced by an estimate $\pi(x)$ and the cost $C_{\exp}(x)$ is replaced by $\hat{C}_{\exp}(x)$:

$$\hat{R}_{\mathrm{def}}(h) = \mathbb{E}_x \left[ \langle \pi(x), e_{h(x)}^{\mathrm{mod}} + \hat{e}_{h(x)}^{\exp}(x) \rangle \right],$$

where we denote $\hat{e}_{\hat{y}}^{\exp}(x) = \left[ \hat{C}_{\exp}(x) \cdot \mathbf{1}(\hat{y} \neq \perp), \ldots, \hat{C}_{\exp}(x) \cdot \mathbf{1}(\hat{y} \neq \perp) \right]^\top \in \mathbb{R}_+^L$.

Finally, consider an empirical version of the Bayes-optimal classifier in (3):

$$\bar{h}(x) = \begin{cases} \underset{y \in \mathcal{Y}}{\mathrm{argmax}}\, \pi_y(x) & \text{if } 1 - \max_{y \in [L]} \pi_y(x) < \hat{C}_{\exp}(x) \\ \perp & \text{else} \end{cases}.$$

From the definition of $\bar{h}$, we have that $\bar{h}(x)$ minimizes $\langle \pi(x), e_{h(x)}^{\mathrm{mod}} + \hat{e}_{h(x)}^{\exp}(x) \rangle$ for each $x$, and as a result, for any classifier $h$:

$$\hat{R}_{\mathrm{def}}(\bar{h}) \leq \hat{R}_{\mathrm{def}}(h). \tag{16}$$

The excess risk can then be expanded as:

$$R_{\mathrm{def}}(\bar{h}) - R_{\mathrm{def}}(\bar{h}^*)$$
$$= R_{\mathrm{def}}(\bar{h}) - \hat{R}_{\mathrm{def}}(\bar{h}) + \hat{R}_{\mathrm{def}}(\bar{h}) - R_{\mathrm{def}}(\bar{h}^*)$$
$$\leq R_{\mathrm{def}}(\bar{h}) - \hat{R}_{\mathrm{def}}(\bar{h}) + \hat{R}_{\mathrm{def}}(\bar{h}^*) - R_{\mathrm{def}}(\bar{h}^*) \tag{17}$$
$$= \mathbb{E}_x \left[ \langle \eta(x), e_{\bar{h}(x)}^{\mathrm{mod}} + e_{\bar{h}(x)}^{\exp}(x) \rangle \right] - \mathbb{E}_x \left[ \langle \pi(x), e_{\bar{h}(x)}^{\mathrm{mod}} + \hat{e}_{\bar{h}(x)}^{\exp}(x) \rangle \right]$$
$$\qquad + \mathbb{E}_x \left[ \langle \pi(x), e_{\bar{h}^*(x)}^{\mathrm{mod}} + \hat{e}_{\bar{h}^*(x)}^{\exp}(x) \rangle \right] - \mathbb{E}_x \left[ \langle \eta(x), e_{\bar{h}^*(x)}^{\mathrm{mod}} + e_{\bar{h}^*(x)}^{\exp}(x) \rangle \right]$$
$$= \mathbb{E}_x \left[ \langle \eta(x), e_{\bar{h}(x)}^{\mathrm{mod}} + e_{\bar{h}(x)}^{\exp}(x) \rangle \right] - \mathbb{E}_x \left[ \langle \pi(x), e_{\bar{h}(x)}^{\mathrm{mod}} + e_{\bar{h}(x)}^{\exp}(x) \rangle \right]$$
$$\qquad + \mathbb{E}_x \left[ \langle \pi(x), e_{\bar{h}^*(x)}^{\mathrm{mod}} + e_{\bar{h}^*(x)}^{\exp}(x) \rangle \right] - \mathbb{E}_x \left[ \langle \eta(x), e_{\bar{h}^*(x)}^{\mathrm{mod}} + e_{\bar{h}^*(x)}^{\exp}(x) \rangle \right]$$
$$\qquad + \mathbb{E}_x \left[ \langle \pi(x), e_{\bar{h}(x)}^{\exp}(x) - \hat{e}_{\bar{h}(x)}^{\exp}(x) \rangle \right] + \mathbb{E}_x \left[ \langle \pi(x), \hat{e}_{\bar{h}^*(x)}^{\exp}(x) - e_{\bar{h}^*(x)}^{\exp}(x) \rangle \right]$$
$$\leq \mathbb{E}_x \left[ \langle \eta(x) - \pi(x), e_{\bar{h}(x)}^{\mathrm{mod}} + e_{\bar{h}(x)}^{\exp}(x) - e_{\bar{h}^*(x)}^{\mathrm{mod}} - e_{\bar{h}^*(x)}^{\exp}(x) \rangle \right]$$
$$\qquad + \mathbb{E}_x \left[ \|\pi(x)\|_1 \cdot \|e_{\bar{h}(x)}^{\exp}(x) - \hat{e}_{\bar{h}(x)}^{\exp}(x)\|_\infty \right] + \mathbb{E}_x \left[ \|\pi(x)\|_1 \cdot \|e_{\bar{h}^*(x)}^{\exp}(x) - \hat{e}_{\bar{h}^*(x)}^{\exp}(x)\|_\infty \right]$$
$$\tag{18}$$
$$\leq \mathbb{E}_x \left[ \langle \eta(x) - \pi(x), e_{\bar{h}(x)}^{\mathrm{mod}} + e_{\bar{h}(x)}^{\exp}(x) - e_{\bar{h}^*(x)}^{\mathrm{mod}} - e_{\bar{h}^*(x)}^{\exp}(x) \rangle \right] + 2 \cdot \mathbb{E}_x \left[ \left| C_{\exp}(x) - \hat{C}_{\exp}(x) \right| \right]$$
$$\tag{19}$$
$$\leq \mathbb{E}_x \left[ \|\eta(x) - \pi(x)\|_1 \cdot \|e_{\bar{h}(x)}^{\mathrm{mod}} + e_{\bar{h}(x)}^{\exp}(x) - e_{\bar{h}^*(x)}^{\mathrm{mod}} - e_{\bar{h}^*(x)}^{\exp}(x)\|_\infty + 2 \left| C_{\exp}(x) - \hat{C}_{\exp}(x) \right| \right],$$
$$\tag{20}$$

where the inequality in (17) follows from (16) with $h = h^*$; the inequality in (18) follows from Hölder's inequality; and the inequality in (19) uses the fact that $\|\pi(x)\|_1 = 1$ and $\mathbf{1}(\bar{h}(x) \neq \perp) \leq 1$. The last step uses Hölder's inequality.

Further expanding the inner $\infty$-norm term in (20), we have for any $x$:

$$\|e_{\bar{h}(x)}^{\text{mod}} + e_{\bar{h}(x)}^{\text{exp}}(x) - e_{\bar{h}^*(x)}^{\text{mod}} - e_{\bar{h}^*(x)}^{\text{exp}}(x)\rangle\|_\infty$$
$$\leq \max_{y \in [L]} \left| \mathbf{1}(\bar{h}(x) \neq y) + C_{\exp}(x) \cdot \mathbf{1}(\bar{h}(x) = \perp) - \mathbf{1}(\bar{h}^*(x) \neq y) - C_{\exp}(x) \cdot \mathbf{1}(\bar{h}^*(x) = \perp) \right|$$
$$\leq \max\{0, 1, 1 - C_{\exp}(x), C_{\exp}(x) - 1, C_{\exp}(x)\}$$
$$\leq \max\{1, C_{\exp}(x)\}$$
$$\leq c_{\max},$$

where we use the fact that $C_{\exp}(x) \leq c_{\max}$ and $c_{\max} \geq 1$. Substituting this back in (20), we get:

$$R_{\text{def}}(\bar{h}) - R_{\text{def}}(\bar{h}^*) \leq c_{\max} \cdot \mathbb{E}_x \left[ \|\eta(x) - \pi(x)\|_1 \right] + 2\mathbb{E}_x \left[ \left| C_{\exp}(x) - \hat{C}_{\exp}(x) \right| \right].$$

Setting $C_{\exp}(x) = \mathbb{P}(y \neq h_{\exp}(x)) + c_0$ and $\hat{C}_{\exp}(x) = \epsilon(x) + c_0$ completes the proof. $\qquad\square$

### A.3 Proof of Lemma 3

*Proof.* The conditional risk for the loss is

$$\mathbb{E}_{y|x} [\ell(x, y, r(x))] = \mathbb{E}_{y|x} [c_{\text{mod}}(x, y)] \cdot \phi(r(x)) + \mathbb{E}_{y|x} [c_{\exp}(x, y)] \cdot \phi(-r(x))$$
$$\propto a(x) \cdot \phi(r(x)) + (1 - a(x)) \cdot \phi(-r(x)),$$

where

$$a(x) \doteq \frac{\mathbb{E}_{y|x} [c_{\text{mod}}(x, y)]}{\mathbb{E}_{y|x} [c_{\text{mod}}(x, y)] + \mathbb{E}_{y|x} [c_{\exp}(x, y)]}.$$

Consequently, since $\phi$ is strictly proper composite, the Bayes-optimal must satisfy [28] $\psi(r^*(x)) = a(x)$. Note that $a(x) > 0.5 \iff \mathbb{E}_{y|x} [c_{\text{mod}}(x, y)] > \mathbb{E}_{y|x} [c_{\exp}(x, y)]$, i.e., the cost of querying the model exceeds that of querying the expert. $\qquad\square$

### A.4 Proof of Theorem 4

Let $\bar{h}^*$ denote the Bayes-optimal classifier that minimizes the L2D risk in (13). We can re-write $\bar{h}^*$ in terms of a base classifier and a rejector:

$$\bar{h}^*(x) = \begin{cases} h^*(x) & \text{if } r^*(x) < 0 \\ \perp & \text{else.} \end{cases},$$

where $h^*(x) \in \operatorname*{argmax}_{y \in \mathcal{Y}} \mathbb{P}(y \mid x)$ and $\text{sign}(r^*(x)) = \text{sign}\left(1 - \max_{y \in \mathcal{Y}} \mathbb{P}(y \mid x) - \mathbb{E}_{y|x} [c_{\exp}(x, y)]\right)$.

Notice that $h^*$ is also the Bayes-optimal classifier for the standard misclassification error $R_{\text{err}}(h)$.

We will find it useful to first define a normalized version of the L2D risk:

$$\tilde{R}_{\text{def}}(h, r) = \mathbb{E}_x \left[ \frac{1}{Z(x)} \cdot L(x, h, r) \right],$$

where

$$L(x, h, r) = \mathbb{E}_{y|x} [\mathbf{1}(y \neq h(x))] \cdot \mathbf{1}(r(x) < 0) + \mathbb{E}_{y|x} [c_{\exp}(x, y)] \cdot \mathbf{1}(r(x) > 0)$$

and

$$Z(x) = \mathbb{E}_{y|x} [\mathbf{1}(y \neq h(x)) + c_{\exp}(x, y)].$$

Similarly, we define the normalized partial surrogate risk as:

$$\tilde{R}_{\text{def}}^\phi(h, r) = \mathbb{E}_x \left[ \frac{1}{Z(x)} \cdot L^\phi(x, h, r) \right].$$

where

$$L^\phi(x, h, r) = \mathop{\mathbb{E}}_{y|x} \left[ \mathbf{1}(y \neq h(x)) \right] \cdot \phi(r(x)) + \mathop{\mathbb{E}}_{y|x} \left[ c_{\exp}(x, y) \right] \cdot \phi(-r(x)).$$

We will also find the following lemmas useful for proving the theorem.

**Lemma 5.** *Suppose $c_{\exp}(x, y) \in [c_{\min}, c_{\max}]$ for some $c_{\min} > 0$. Then for any fixed base classifier $\hat{h}$, the normalized excess risks can be bounded in terms of the unnormalized excess risks as follows:*

$$\tilde{R}_{\mathrm{def}}(\hat{h}, \hat{r}) - \min_{r:\mathcal{X}\to\mathbb{R}} \tilde{R}_{\mathrm{def}}(\hat{h}, r) \geq \frac{1}{1 + c_{\max}} \cdot \left( R_{\mathrm{def}}(\hat{h}, \hat{r}) - \min_{r:\mathcal{X}\to\mathbb{R}} R_{\mathrm{def}}(\hat{h}, r) \right),$$

$$\tilde{R}^\phi_{\mathrm{def}}(\hat{h}, \hat{r}) - \min_{r:\mathcal{X}\to\mathbb{R}} \tilde{R}^\phi_{\mathrm{def}}(\hat{h}, r) \leq \frac{1}{c_{\min}} \cdot \left( R^\phi_{\mathrm{def}}(\hat{h}, \hat{r}) - \min_{r:\mathcal{X}\to\mathbb{R}} R^\phi_{\mathrm{def}}(\hat{h}, r) \right).$$

*Proof.* Expanding the normalized excess misclassification risk, we have:

$$\tilde{R}_{\mathrm{def}}(\hat{h}, \hat{r}) - \min_{r:\mathcal{X}\to\mathbb{R}} \tilde{R}_{\mathrm{def}}(\hat{h}, r) = \mathbb{E}_x \left[ \frac{1}{Z(x)} \cdot L(x, \hat{h}, \hat{r}) \right] - \min_{r:\mathcal{X}\to\mathbb{R}} \mathbb{E}_x \left[ \frac{1}{Z(x)} \cdot L(x, \hat{h}, r) \right]$$

$$= \mathbb{E}_x \left[ \frac{1}{Z(x)} \cdot L(x, \hat{h}, \hat{r}) \right] - \mathbb{E}_x \left[ \frac{1}{Z(x)} \cdot \min_{z \in \mathbb{R}} L(x, \hat{h}, z) \right]$$

$$= \mathbb{E}_x \left[ \frac{1}{Z(x)} \cdot \left( L(x, \hat{h}, \hat{r}) - \min_{z \in \mathbb{R}} L(x, \hat{h}, z) \right) \right]$$

$$\geq \mathbb{E}_x \left[ \frac{1}{1 + c_{\max}} \cdot \left( L(x, \hat{h}, \hat{r}) - \min_{z \in \mathbb{R}} L(x, \hat{h}, z) \right) \right]$$

$$= \frac{1}{1 + c_{\max}} \cdot \left( R_{\mathrm{def}}(\hat{h}, \hat{r}) - \min_{r:\mathcal{X}\to\mathbb{R}} R_{\mathrm{def}}(\hat{h}, r) \right),$$

where in the second step, we use the fact that the optimal rejector is point-wise optimal, and in the penultimate step, we use $Z(x) = \mathop{\mathbb{E}}_{y|x} \left[ \mathbf{1}(y \neq h(x)) + c_{\exp}(x, y) \right] \leq 1 + c_{\max}$.

Similarly, expanding the normalized excess surrogate risk, we have:

$$\tilde{R}^\phi_{\mathrm{def}}(\hat{h}, \hat{r}) - \min_{r:\mathcal{X}\to\mathbb{R}} \tilde{R}^\phi_{\mathrm{def}}(\hat{h}, r) = \mathbb{E}_x \left[ \frac{1}{Z(x)} \cdot L^\phi(x, \hat{h}, \hat{r}) \right] - \min_{r:\mathcal{X}\to\mathbb{R}} \mathbb{E}_x \left[ \frac{1}{Z(x)} \cdot L^\phi(x, \hat{h}, r) \right]$$

$$= \mathbb{E}_x \left[ \frac{1}{Z(x)} \cdot \left( L(x, \hat{h}, \hat{r}) - \min_{z \in \mathbb{R}} L(x, \hat{h}, z) \right) \right]$$

$$\leq \frac{1}{c_{\min}} \cdot \left( R^\phi_{\mathrm{def}}(\hat{h}, \hat{r}) - \min_{r:\mathcal{X}\to\mathbb{R}} R^\phi_{\mathrm{def}}(\hat{h}, r) \right),$$

where we have used the fact $Z(x) \geq c_{\min}$. $\qquad\square$

The next lemma builds on Lemma 5 to provide a excess risk bound for the rejector.

**Lemma 6.** *Suppose $\phi$ is classification calibrated and $c_{\exp}(x, y) \in [c_{\min}, c_{\max}]$ for some $c_{\min} > 0$. Then for any fixed base classifier $\hat{h}$,*

$$R_{\mathrm{def}}(\hat{h}, \hat{r}) - \min_{r:\mathcal{X}\to\mathbb{R}} R_{\mathrm{def}}(\hat{h}, r) \leq \Psi \left( R^\phi_{\mathrm{def}}(\hat{h}, \hat{r}) - \min_{r:\mathcal{X}\to\mathbb{R}} R^\phi_{\mathrm{def}}(\hat{h}, r) \right),$$

*for some increasing function $\Psi : \mathbb{R}_+ \to \mathbb{R}_+$ with $\Psi(0) = 0$.*

*Proof.* Using the fact that $\phi$ is classification calibrated, one can directly apply the surrogate excess risk bounds from Bartlett et al. [3] to the normalized surrogate risk. Specifically, we first note that for the label distribution $q(x) = \frac{1}{Z(x)} \cdot \mathop{\mathbb{E}}_{y|x} \left[ \mathbf{1}(y \neq h(x)) \right]$, the normalized risks can be written as:

$$\tilde{R}_{\mathrm{def}}(h, r) = q(x) \cdot \phi(r(x)) + (1 - q(x)) \cdot \phi(-r(x));$$

$$\tilde{R}_{\text{def}}^{\phi}(h, r) = q(x) \cdot \mathbf{1}(r(x) < 0) + (1 - q(x)) \cdot \mathbf{1}(r(x) > 0).$$

We can then bound the excess normalized surrogate risk in terms of the excess normalized misclassification risk:

$$\tilde{R}_{\text{def}}(\hat{h}, \hat{r}) - \min_{r:\mathcal{X} \to \mathbb{R}} \tilde{R}_{\text{def}}(\hat{h}, r) \leq \zeta \left( \tilde{R}_{\text{def}}^{\phi}(\hat{h}, \hat{r}) - \min_{r:\mathcal{X} \to \mathbb{R}} \tilde{R}_{\text{def}}^{\phi}(\hat{h}, r) \right),$$

for some increasing function $\zeta : \mathbb{R}_+ \to \mathbb{R}_+$ with $\zeta(0) = 0$.

We then apply Lemma 5 to lower bound the LHS and upper bound the RHS:

$$\frac{1}{1 + c_{\max}} \cdot \left( R_{\text{def}}(\hat{h}, \hat{r}) - \min_{r:\mathcal{X} \to \mathbb{R}} R_{\text{def}}(\hat{h}, r) \right) \leq \zeta \left( \frac{1}{c_{\min}} \cdot \left( R_{\text{def}}^{\phi}(\hat{h}, \hat{r}) - \min_{r:\mathcal{X} \to \mathbb{R}} R_{\text{def}}^{\phi}(\hat{h}, r) \right) \right)$$

from which we have:

$$\left( R_{\text{def}}(\hat{h}, \hat{r}) - \min_{r:\mathcal{X} \to \mathbb{R}} R_{\text{def}}(\hat{h}, r) \right) \leq (1 + c_{\max}) \cdot \zeta \left( \frac{1}{c_{\min}} \cdot \left( R_{\text{def}}^{\phi}(\hat{h}, \hat{r}) - \min_{r:\mathcal{X} \to \mathbb{R}} R_{\text{def}}^{\phi}(\hat{h}, r) \right) \right).$$

Setting $\Psi(z) = (1 + c_{\max}) \cdot \zeta \left( \frac{1}{c_{\min}} \cdot z \right)$ completes the proof. $\qquad\square$

We are now ready to prove Theorem 4.

*Proof of Theorem 4.* We have:

$$R_{\text{def}}(\hat{h}, \hat{r}) - R_{\text{def}}(h^*, r^*) = \underbrace{R_{\text{def}}(\hat{h}, \hat{r}) - \min_{r:\mathcal{X} \to \mathbb{R}} R_{\text{def}}(\hat{h}, r)}_{\text{term}_1} + \underbrace{\min_{r:\mathcal{X} \to \mathbb{R}} R_{\text{def}}(\hat{h}, r) - R_{\text{def}}(h^*, r^*)}_{\text{term}_2}.$$

$$(21)$$

We first bound the second term below:

$$\min_{r:\mathcal{X} \to \mathbb{R}} R_{\text{def}}(\hat{h}, r) - R_{\text{def}}(h^*, r^*) \leq R_{\text{def}}(\hat{h}, r^*) - R_{\text{def}}(h^*, r^*)$$

$$\leq \mathbb{E} \left[ \mathbf{1}(r^*(x) < 0) \cdot \left( \mathbf{1}(y \neq \hat{h}(x)) - \mathbf{1}(y \neq h^*(x)) \right) \right]$$

$$= \mathbb{E}_x \left[ \mathbf{1}(r^*(x) < 0) \cdot \left( \mathbb{E}_{y|x} \left[ \mathbf{1}(y \neq \hat{h}(x)) \right] - \mathbb{E}_{y|x} \left[ \mathbf{1}(y \neq h^*(x)) \right] \right) \right]$$

$$\leq \mathbb{E}_x \left[ \mathbb{E}_{y|x} \left[ \mathbf{1}(y \neq \hat{h}(x)) \right] - \mathbb{E}_{y|x} \left[ \mathbf{1}(y \neq h^*(x)) \right] \right].$$

where the last step uses the fact that $\mathbf{1}(r^*(x) < 0) \leq 1$ and by definition of $h^*$, $\mathbb{E}_{y|x} \left[ \mathbf{1}(y \neq h(x)) \right] \geq \mathbb{E}_{y|x} \left[ \mathbf{1}(y \neq h^*(x)) \right]$, for all classifiers $h$.

Substituting this back into (21) to bound the second term, and applying Lemma 6 to bound the first term, we have:

$$R_{\text{def}}(\hat{h}, \hat{r}) - R_{\text{def}}(h^*, r^*)$$

$$\leq \Psi \left( R_{\text{def}}^{\phi}(\hat{h}, \hat{r}) - \min_{r:\mathcal{X} \to \mathbb{R}} R_{\text{def}}^{\phi}(\hat{h}, r) \right) + \mathbb{E} \left[ \mathbf{1}(y \neq \hat{h}(x)) \right] - \mathbb{E} \left[ \mathbf{1}(y \neq h^*(x)) \right]$$

$$= \Psi \left( R_{\text{def}}^{\phi}(\hat{h}, \hat{r}) - \min_{r:\mathcal{X} \to \mathbb{R}} R_{\text{def}}^{\phi}(\hat{h}, r) \right) + \mathbb{E} \left[ \mathbf{1}(y \neq \hat{h}(x)) \right] - \min_{h:\mathcal{X} \to [L]} \mathbb{E} \left[ \mathbf{1}(y \neq h^*(x)) \right],$$

where $\Psi$ is some increasing function with $\Psi(0) = 0$, and in the second inequality, we use the fact that $h^*$ is the Bayes-optimal classifier for the misclassification error. $\qquad\square$

## B   Learning to defer as cost-sensitive learning

For a classifier $h: \mathcal{X} \to \mathcal{Y}$, the misclassification error $R(h) = \mathbb{P}(y \neq h(x))$ assumes that all mispredictions are equally undesirable, and that the classifier can only output labels from $\mathcal{Y}$. More generally, *cost-sensitive classification* [11] seeks $h: \mathcal{X} \to \hat{\mathcal{Y}}$ that minimises

$$R_{\text{cs}}(h) = \mathbb{E}_{(x,y)} \left[ \alpha(x, y, h(x)) \right],$$

for *cost function* $\alpha\colon \mathcal{X}\times\mathcal{Y}\times\hat{\mathcal{Y}}\to\mathbb{R}_+$ and *prediction space* $\hat{\mathcal{Y}}$. When $\hat{\mathcal{Y}}=\mathcal{Y}$ and $\alpha(x,y,\hat{y})=\mathbf{1}(y\neq\hat{y})$, $R_{\text{cs}}(h)=R(h)$. The Bayes-optimal cost-sensitive classifier is

$$(\forall x\in\mathcal{X})\, h^*(x)=\operatorname*{argmin}_{\hat{y}\in\hat{\mathcal{Y}}}\; \mathbb{E}_{y|x}\left[\alpha(x,y,\hat{y})\right]. \tag{22}$$

In the *learning to defer* (*L2D*) problem [18], one seeks a classifier that can either make a standard prediction in $\mathcal{Y}$, or *defer* its prediction to an *expert* model $h_{\text{exp}}\colon\mathcal{X}\to\mathcal{Y}$. Invoking an expert carries an associated sample-dependent *cost* $c_{\text{exp}}(x,y)>0$, so as to prevent the classifier from deferring on all samples. Concretely, let $\hat{\mathcal{Y}}=\mathcal{Y}\cup\{\perp\}$, where $\perp$ denotes the "defer" option. Consider a classifier $\bar{h}\colon\mathcal{X}\to\hat{\mathcal{Y}}$ equipped with a "defer" option $\perp$. Our goal is then to minimise

$$R_{\text{def}}(\bar{h})=\mathbb{E}_{(x,y)}\left[\alpha(x,y,\bar{h}(x))\right]\ \text{for}\ \alpha(x,y,\hat{y})=\begin{cases}\mathbf{1}(y\neq\hat{y}) & \text{if } \hat{y}\neq\perp\\ c_{\text{exp}}(x,y) & \text{else.}\end{cases} \tag{23}$$

## C   On the Bayes-optimal deferral logit and probability

We now detail the form of the Bayes-optimal defer logit $\bar{f}^*_\perp(x)$ for the cost-sensitive softmax cross-entropy, assuming that all other logits $\bar{f}_1(x),\dots,\bar{f}_L(x)$ are fixed, and chosen arbitrarily. This is slightly more general than (5), which considered the joint Bayes-optimal solution for *all* logits. This analysis highlights two points: first, even when the other logits $\bar{f}_1(x),\dots,\bar{f}_L(x)$ are chosen arbitrarily, there is no dependence of the optimal defer *probability* $\bar{p}^*_\perp(x)$ on the logits $\bar{f}_1(x),\dots,\bar{f}_L(x)$. Second, as a qualifier to the above, there is however a dependence of the optimal defer *logit* $\bar{f}^*_\perp(x)$ on the logits $\bar{f}_1(x),\dots,\bar{f}_L(x)$.

From (4), the cost-sensitive softmax cross-entropy has conditional risk

$$L(x,\bar{f}(x))=\mathbb{E}_{y|x}\left[\ell_{\text{CSS}}(x,y,\bar{f}(x))\right]$$

$$=\mathbb{E}_{y|x}\left[\sum_{y'\in\mathcal{Y}}(c_{\max}-\mathbf{1}(y\neq y'))\cdot-\log\bar{p}_{y'}(x)\right]+\mathbb{E}_{y|x}\left[(c_{\max}-c_{\text{exp}}(x,y))\cdot-\log\bar{p}_\perp(x)\right]$$

$$=\sum_{y'\in\mathcal{Y}}\mathbb{E}_{y|x}\left[(c_{\max}-\mathbf{1}(y\neq y'))\right]\cdot-\log\bar{p}_{y'}(x)+\mathbb{E}_{y|x}\left[(c_{\max}-c_{\text{exp}}(x,y))\right]\cdot-\log\bar{p}_\perp(x)$$

$$=\sum_{y'\in\mathcal{Y}}(c_{\max}-(1-\mathbb{P}(y'\mid x)))\cdot-\log\bar{p}_{y'}(x)+(c_{\max}-\mathbb{E}_{y|x}\left[c_{\text{exp}}(x,y)\right])\cdot-\log\bar{p}_\perp(x)$$

$$=\sum_{y'\in\mathcal{Y}}(c_{\max}-(1-\mathbb{P}(y'\mid x)))\cdot-\log\frac{\exp(\bar{f}_{y'}(x))}{\bar{Z}(x)}$$

$$+(c_{\max}-\mathbb{E}_{y|x}\left[c_{\text{exp}}(x,y)\right])\cdot-\log\frac{\exp(\bar{f}_\perp(x))}{\bar{Z}(x)},$$

where $\bar{Z}(x)\doteq Z(x)+\exp(\bar{f}_\perp(x))$ and $Z(x)\doteq\sum_{y'\in\mathcal{Y}}\exp(\bar{f}_{y'}(x))$. Thus, the terms involving $\bar{f}_\perp(x)$ are

$$\bar{L}(x,\bar{f}_\perp(x))$$

$$=\left[\sum_{y'\in\mathcal{Y}}(c_{\max}-(1-\mathbb{P}(y'\mid x)))\right]\cdot\log\left[\bar{Z}(x)\right]+(c_{\max}-\mathbb{E}_{y|x}\left[c_{\text{exp}}(x,y)\right])\cdot-\log\frac{\exp(\bar{f}_\perp(x))}{\bar{Z}(x)}$$

$$=\left[L\cdot(c_{\max}-1)+1\right]\cdot\log\left[\bar{Z}(x)\right]+(c_{\max}-\mathbb{E}_{y|x}\left[c_{\text{exp}}(x,y)\right])\cdot-\log\frac{\exp(\bar{f}_\perp(x))}{\bar{Z}(x)}$$

$$=\left[L\cdot(c_{\max}-1)+1+c_{\max}-\mathbb{E}_{y|x}\left[c_{\text{exp}}(x,y)\right]\right]\cdot\log\left[Z(x)+\exp(\bar{f}_\perp(x))\right]$$

$$-(c_{\max}-\mathbb{E}_{y|x}\left[c_{\text{exp}}(x,y)\right])\cdot\bar{f}_\perp(x).$$

At optimality, for any choice of $\bar{f}_1, \ldots, \bar{f}_L$, we thus have

$$\left[ L \cdot (c_{\max} - 1) + 1 + c_{\max} - \mathop{\mathbb{E}}_{y|x} \left[ c_{\exp}(x, y) \right] \right] \cdot \frac{\exp(\bar{f}_{\perp}^*(x))}{Z(x) + \exp(\bar{f}_{\perp}^*(x))} = c_{\max} - \mathop{\mathbb{E}}_{y|x} \left[ c_{\exp}(x, y) \right]$$

$$\iff \bar{p}_{\perp}^*(x) = \frac{c_{\max} - \mathop{\mathbb{E}}_{y|x} \left[ c_{\exp}(x, y) \right]}{L \cdot (c_{\max} - 1) + 1 + c_{\max} - \mathop{\mathbb{E}}_{y|x} \left[ c_{\exp}(x, y) \right]}.$$

Thus, for *any* $\bar{f}_1, \ldots, \bar{f}_L$, the *probability* assigned to the deferral label will be independent of these logits. Note however that the deferral *logit* $\bar{f}_{\perp}^*$ itself will necessarily depend on these logits: intuitively, it serves to re-normalise the softmax probabilities of these logits to ensure $\bar{p}_{\perp}$ takes the desired form. Indeed, some simple algebra reveals that

$$\bar{f}_{\perp}^*(x) = \log Z(x) + \log \frac{c_{\max} - \mathop{\mathbb{E}}_{y|x} \left[ c_{\exp}(x, y) \right]}{L \cdot (c_{\max} - 1) + 1}$$

$$= \log \left[ \sum_{y' \in \mathcal{Y}} \exp(\bar{f}_{y'}(x)) \right] + \log \frac{c_{\max} - \mathop{\mathbb{E}}_{y|x} \left[ c_{\exp}(x, y) \right]}{L \cdot (c_{\max} - 1) + 1}. \tag{24}$$

Thus, the Bayes-optimal defer logit has a non-linear dependence on the logits $\bar{f}_1, \ldots, \bar{f}_L$. We further remark that when the model class is $\bar{f}$ is of insufficiently high capacity, it may not be feasible to achieve this Bayes-optimal solution (even though $\bar{p}_{\perp}^*(x)$ is relatively simple). For example, suppose we parameterise $\bar{f}_{y'}(x) = w_{y'}^\top \Phi(x) + b_{y'}$. Then, it is not possible to express $\bar{f}_{\perp}^*(x)$ from (24) in the form $w_{\perp}^\top \Phi(x) + b_{\perp}$, since $\log \left[ \sum_{y' \in \mathcal{Y}} \exp(\bar{f}_{y'}(x)) \right]$ is non-linear in $\Phi(x)$.

As a final remark, we note that the picture changes when considering the one-versus-all loss: here, regardless of the choice of $\bar{f}_1, \ldots, \bar{f}_L$ the Bayes-optimal solution $\bar{f}_{\perp}^*$ will be $\psi^{-1} \left( 1 - \mathop{\mathbb{E}}_{y|x} \left[ c_{\exp}(x, y) \right] \right)$. This is owing to the decoupled nature of the loss; consequently, there is no influence of the other logits on $\bar{f}_{\perp}^*$.

## D  General one-versus-all loss

We now present a more general form of the one-versus-all loss, where we allow for a generic cost function $\alpha(x, y, \bar{h}(x))$ following (23). Consider a loss of the form

$$\ell(x, y, \bar{f}(x)) = \sum_{y' \in \hat{\mathcal{Y}}} \left[ a_{yy'}(x) \cdot \phi(f_y(x)) + b_{yy'}(x) \cdot \phi(f_{y'}(x)) \right] \tag{25}$$

for $a_{yy'}(x), b_{yy'}(x) \in \mathbb{R}$. The conditional risk is

$$\mathop{\mathbb{E}}_{y|x} \left[ \ell(x, y, \bar{f}(x)) \right] = \sum_{y' \in \hat{\mathcal{Y}}} \left[ A_{y'}(x) \cdot \phi(f_y(x)) + B_{y'}(x) \cdot \phi(f_{y'}(x)) \right],$$

where $A_{y'}(x) \doteq \mathop{\mathbb{E}}_{y|x} \left[ a_{yy'}(x) \right]$ and $B_{y'}(x) \doteq \mathop{\mathbb{E}}_{y|x} \left[ b_{yy'}(x) \right]$. The Bayes-optimal scorer for a strictly proper composite loss with inverse link $\psi$ is thus

$$(\forall x \in \mathcal{X}) \, \psi(f_{y'}^*(x)) = \frac{A_{y'}(x)}{B_{y'}(x) + B_{y'}(x)} \tag{26}$$

Now observe that the Bayes-optimal solution for the cost-sensitive problem (22) is expressible as

$$h^*(x) = \underset{\hat{y} \in \hat{\mathcal{Y}}}{\operatorname{argmax}} \, f_{y'}^*(x)$$

$$f_{y'}^*(x) = 1 - \frac{\mathop{\mathbb{E}}_{y|x} \left[ \alpha(x, y, y') \right]}{\alpha_{\max}},$$

where $\alpha_{\max}$ is an upper bound on $\alpha(x, y, y')$. Thus, if we set

$$a_{yy'}(x) = 1 - \frac{\alpha(x, y, y')}{\alpha_{\max}}$$

$$b_{yy'}(x) = \frac{\alpha(x, y, y')}{\alpha_{\max}},$$

the Bayes-optimal scorer in (26) will precisely yield $\psi(f_{y'}^*(x)) = 1 - \frac{\mathbb{E}_{y|x}\left[\alpha(x, y, y')\right]}{\alpha_{\max}}$.

As a sanity check, when $\alpha$ corresponds to the learning to defer risk, we have

$$a_{yy'}(x) = \begin{cases} \frac{c_{\max} - \mathbf{1}(y \neq y')}{c_{\max}} & \text{if } y' \neq \perp \\ \frac{c_{\max} - c_{\exp}(x, y)}{c_{\max}} & \text{if } y' = \perp \end{cases}$$

$$b_{yy'}(x) = \begin{cases} \frac{\mathbf{1}(y \neq y')}{c_{\max}} & \text{if } y' \neq \perp \\ \frac{c_{\exp}(x, y)}{c_{\max}} & \text{if } y' = \perp, \end{cases}$$

which is seen to yield the OvA loss in (10).

## E    Summary of losses for learning to defer

Table 1 summarises various approaches for learning to defer. Each method employs a base loss $\ell(y, x, f(x); c_{\text{base}})$ to train the base model with a base deferral cost $c_{\text{base}}$, and optionally optimizes a rejector model using the rejector loss provided in (14) with the user-specified fixed deferral cost $c_0$:

$$\ell_{\text{rej}}(x, y, r(x); c_0) = c_{\text{mod}}(x, y) \cdot \phi(r(x)) + (c_0 + \mathbf{1}\left(y \neq h_{\exp}(x)\right)) \cdot \phi(-r(x)).$$

Prior methods by Mozannar and Sontag [19] and Verma and Nalisnick [34] include the user-specified cost $c_0$ in the base loss, i.e., set $c_{\text{base}} = c_0$, and do not train an explicit rejector. The proposed methods in this paper set $c_{\text{base}} = 0$ in the base loss, and include $c_0$ either as a part of a *post-hoc thresholding* step, or as a part of the rejector loss in an explicit *post-hoc training* step.

We also include a variant where one minimizes the standard softmax cross-entropy loss $\ell_{\text{CE}}(y, f(x)) = \log\left[\sum_{y' \in \mathcal{Y}} e^{f_{y'}(x)}\right] - f_y(x)$ to train the base model, and trains the rejector in a post-hoc manner.

## F    Parameterising the post-hoc rejector

In constructing the rejector model $r : \mathcal{X} \to \mathbb{R}$, it may be beneficial to paramterize it in terms of the base model's probabilities. For example, if the base loss is the CSS loss in (8), then we may want to use a rejector of the form:

$$r(x) = g_\perp(x) + \frac{\bar{p}_\perp(x)}{1 - \bar{p}_\perp(x)} - \max_{y \in [L]} \frac{\bar{p}_y(x)}{1 - \bar{p}_\perp(x)},$$

where $\bar{p}$ denotes a softmax transformation of the $L + 1$ logits from the fixed base model $\bar{f}$, and $g_\perp : \mathcal{X} \to \mathbb{R}$ is a example-dependent bias term that we explicitly train.

Notice that if the first $L$ probabilities from the base model accurately estimate the underlying conditional-class probabilities, i.e., $\bar{p}_y(x) \propto \mathbb{P}(y \mid x), \forall y \in [L]$, and the reject probability is an accurate estimate of the expert's error, i.e., $\bar{p}_\perp(x) \propto \mathbb{P}(y \neq h_{\exp}(x))$, then $r(x) > 0$ matches the Bayes-optimal deferral decision when $g_\perp(x) = c_0$. In practice, the base model may not accurately estimate either $\mathbb{P}(y \mid x)$ or the expert's error, and therefore the additional term $g_\perp(x)$ that we train can be seen as correcting errors in the base model's estimates.

Table 1 also presents the rejector parameterisation for other base losses, such as the OvA loss and the SOvA loss. In each case, the form of the rejector is chosen to recover the Bayes-optimal decision rule when the base model provides calibrated probability estimates for $\mathbb{P}(y \mid x)$ and $\mathbb{P}(y \neq h_{\exp}(x))$. We employ these parameterisations in all our experiments in §5.

Table 1: Summary of old and new approaches for learning to defer. We use $\bar{f}(x)$ and $f(x)$ to denote scoring functions with and without defer option $\perp$. We use $\bar{p}(x)$ to denote a softmax transformation on all $L + 1$ logits of $\bar{f}$, and $p(x)$ to denote a softmax transformation on the first $L$ logits of either $f$ or $\bar{f}$. We use $\sigma(z) = \frac{1}{1+e^{-z}}$ to denote the sigmoid function. Rows 2–5 describe the post-hoc thresholding approach in §4.1. Rows 6–8 describe the post-hoc training approach in §4.2.

| Base Loss | Rejector Loss | Deferral rule | Description |
|---|---|---|---|
| $\ell_{\mathrm{CSS}}(y, x, \bar{f}(x); c_0)$ | — | $\max_y \bar{p}_y(x) < \bar{p}_\perp(x)$ | Cost-sensitive CE [19] |
| $\ell_{\mathrm{OVA}}(y, x, \bar{f}(x); c_0)$ | — | $\max_y \bar{f}_y(x) < \bar{f}_\perp(x)$ | Cost-sensitive OVA [34] |
| $\ell_{\mathrm{CSS}}(y, x, \bar{f}(x); 0)$ | — | $\max_y \bar{p}_y(x) < (1 + c_0) \cdot \bar{p}_\perp(x) - c_0$ | CSS + Post-thresholding |
| $\ell_{\mathrm{OVA}}(y, x, \bar{f}(x); 0)$ | — | $\max_y \sigma\left(\bar{f}_y(x)\right) < \sigma\left(\bar{f}_\perp(x)\right) - c_0$ | OVA + Post-thresholding |
| $\ell_{\mathrm{SOVA}}(y, x, \bar{f}(x); 0)$ | — | $\max_y p_y(x) < \sigma\left(\bar{f}_\perp(x)\right) - c_0$ | SOVA + Post-thresholding |
| $\ell_{\mathrm{CE}}(y, f(x))$ | $\ell_{\mathrm{rej}}(y, x, r(x); c_0)$, where $r(x) = g_\perp(x) - \max_{y \in [L]} p_y(x)$ | $r(x) > 0$ | CE + Post-hoc training |
| $\ell_{\mathrm{CSS}}(y, x, \bar{f}(x); 0)$ | $\ell_{\mathrm{rej}}(y, x, r(x); c_0)$, where $r(x) = g_\perp(x) + \frac{\bar{p}_\perp(x)}{1 - \bar{p}_\perp(x)} - \max_{y \in [L]} \frac{\bar{p}_y(x)}{1 - \bar{p}_\perp(x)}$ | $r(x) > 0$ | CSS + Post-hoc training |
| $\ell_{\mathrm{OVA}}(y, x, \bar{f}(x); 0)$ | $\ell_{\mathrm{rej}}(y, x, r(x); c_0)$, where $r(x) = g_\perp(x) + \sigma\left(\bar{f}_\perp(x)\right) - \max_{y \in [L]} \sigma\left(\bar{f}_y(x)\right)$ | $r(x) > 0$ | OVA + Post-hoc training |
| $\ell_{\mathrm{SOVA}}(y, x, \bar{f}(x); 0)$ | $\ell_{\mathrm{rej}}(y, x, r(x); c_0)$, where $r(x) = g_\perp(x) + \sigma\left(\bar{f}_\perp(x)\right) - \max_{y \in [L]} p_y(x)$ | $r(x) > 0$ | SOVA + Post-hoc training |

# G   Relation to differentiable triage paradigm

Okati et al. [22] consider a variant of the L2D setup where one is given a *budget* on the fraction of examples on which the model is allowed to defer. The goal is to now learn a base model and deferral rule satisfying this budget constraint. An analogous constrained setting has also been considered in the adaptive inference literature [15]. One potential approach to incorporating post-hoc techniques into this framework is to formulate an equivalent Lagrangian saddle-point problem, and perform alternating *stochastic* updates on $\bar{f}$, and updates on the multiplier associated with the budget constraint. Further exploring this would be of interest.

# H   Details of experiment parameters

## H.1   Synthetic dataset

We construct a distribution $\mathbb{P}(x, y) = \sum_{a \in \{0,1\}} \mathbb{P}(x, y, a)$ as follows. We fix $\mathbb{P}(a = 1) = 0.5$, $\mathbb{P}(y = 1 \mid a) = 0.5$. For subgroup $a = 0$, we set $\mathbb{P}(x \mid y, a = 0) = \mathcal{N}(\mu^{(y,a)}, \Sigma^{(y,a)})$. We draw the elements of $\mu_0^{(0,0)} \sim \mathrm{Unif}(-2, -1)$, $\mu_1^{(0,0)} \sim \mathrm{Unif}(-1, -1)$, and set $\mu_0^{(1,0)} = \mu_0^{(0,0)} + z$ for $z \sim \mathrm{Unif}(3, 4)$. We set $\Sigma^{(0,0)} = \Sigma^{(1,0)}$ to be the identity matrix scaled by a random constant $s \sim \mathrm{Unif}(0, 1)$. Observe that the Gaussians for $a = 0$ both have identical isotropic covariance. The Bayes-optimal solution for the subgroup $\mathbb{P}(x, y \mid a = 0)$ is a linear classifier over the raw features $x$.

For subgroup $a = 1$, we draw $\theta \sim \mathrm{Unif}(0, 2\pi)$, and set $r = 5 + y$. We then collect samples of the form $(r \cos \theta, r \sin \theta)$.

## H.2   Real-world datasets

For all datasets, we perform minibatch SGD optimisation with momentum 0.9. Dataset specific settings are given below.

| Quantity | CIFAR-* | ImageNet |
|---|---|---|
| Weight decay | $10^{-4}$ | $5 \times 10^{-4}$ |
| Epochs | 256 | 90 |
| Batch size | 1024 | 512 |
| Learning rate: peak | 1.0 | 0.4 |
| Learning rate: warmup epochs | 15 | 5 |
| Learning rate: annealing | Decay by $0.1$ after 96th, 192nd, 224th epoch | Cosine |
| Data augmentation | [14] | [12] |

Table 2: Details of hyperparameters for various datasets.

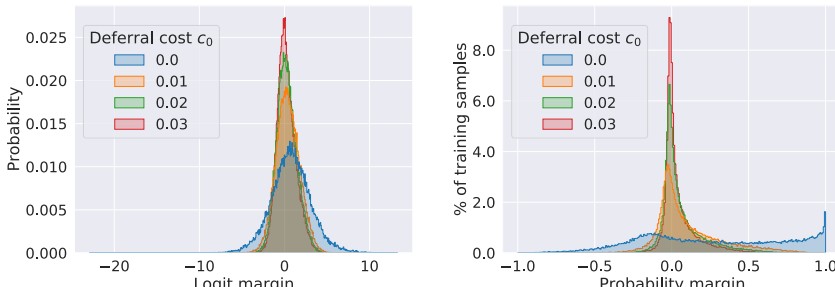

Figure 7: Illustration of underfitting of the cost-sensitive softmax cross-entropy (CSS) loss of Mozannar and Sontag [19] on CIFAR-100. For the setup in Figure 1, we plot the margins in logit and probability space as $c_0$ is varied. These confirm that the range of scores shrinks as $c_0$ increases.

# I    Additional experiments: underfitting of existing losses

## I.1    Underfitting of cost-sensitive softmax cross-entropy

For the setting in Figure 1, Figure 7 shows plots of the *logit margins* $f_y(x) - \max_{y' \neq y} f_{y'}(x)$, and the *probability* margins $p_y(x) - \max_{y' \neq y} p_{y'}(x)$. Both of these degrade as $c_0$ increases, indicating that the probability of the true label is increasingly confusable with that of other labels.

## I.2    Underfitting of one-versus-all loss

For the setting in Figure 1, Figure 7 shows analogous plots for the one-versus-all loss of Verma and Nalisnick [34]. From the top left panel, we see that as $c_0$ increases, the training accuracy degrades. There is a sharper transition here in the margin distributions as $c_0$ becomes non-zero. Nonetheless, we again see that while $c_0 = 0$ produces highly confident predictions on the training set (e.g., from the entropy plot), the introduction of $c_0 > 0$ makes the predictions much less confident (e.g., as evident from the entropy and margin plots).

## I.3    Impact of expert cost function on underfitting

We emphasise here that the underfitting issues above are a consequence of incorporating a fixed deferral cost $c_0 > 0$ in addition to the expert error probability. In settings where $c_0 = 0$, *or* where the expert is assumed to be perfect, such an issue is not present. The former has been considered as part of the plots in the previous section. The latter is potentially applicable in some settings, and indeed corresponds to the classic learning to reject setting [2, 9, 6]. Here, we may set $c_{\max} = 1$, and the cost-sensitive softmax cross-entropy simplifies to

$$\ell_{\text{CSS}}(x, y, \bar{f}(x)) = -\log(\bar{p}_y(x)) - (1 - c_0) \cdot \log(\bar{p}_\perp(x)),$$

which is similar in nature to (8). As with (8), we will not exhibit severe underfitting as a result of the loss only considering $\bar{p}_y$ and $\bar{p}_\perp$, as opposed to all possible labels.

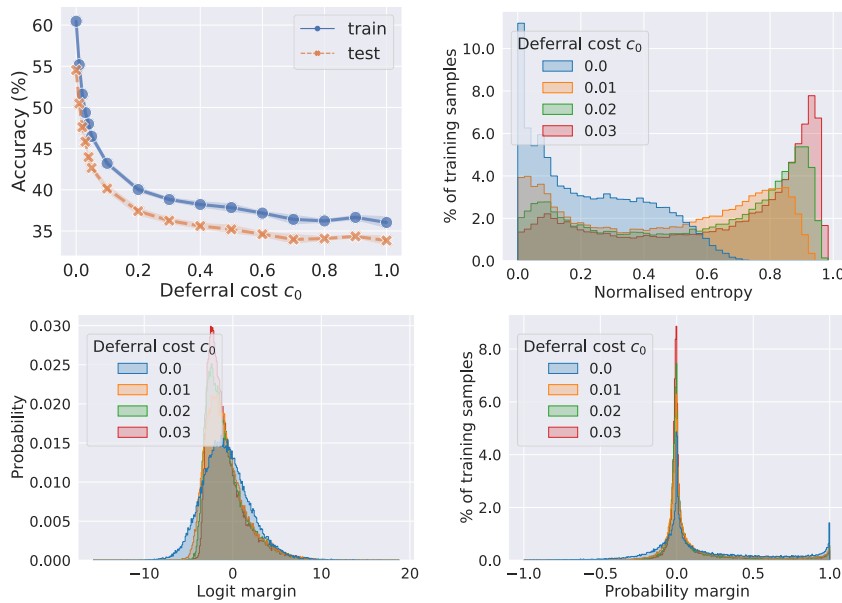

Figure 8: Illustration of underfitting of the **one-versus-all (OvA)** loss of Verma and Nalisnick [34] on CIFAR-100. These confirm that as with the CSS loss, the model underfits as $c_0$ increases, which is a consequence of shrinking margins and increased entropy.

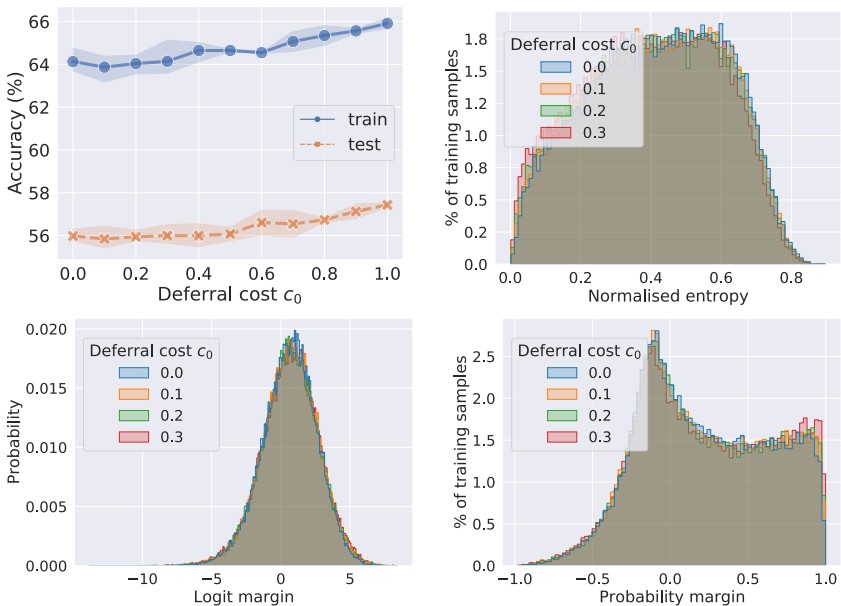

Figure 9: Illustration of impact of choice of expert cost function on underfitting. We follow the same setup as Figure 1 on CIFAR-100, but employ a **constant expert cost** $c_{\exp}(x, y) = c_0 \leq 1$, with $c_{\max} = 1$. Here, we see that as $c_0$ increases, there is **no degradation** of training accuracy. This is owing to this choice of cost only corresponding to a mild form of label smoothing.

To confirm this, we repeat the CIFAR-100 experiment considered in Figure 1 with a constant cost of $c_{\exp}(x, y) = c_0$.

### I.4 Impact of number of labels on underfitting

Figure 10 shows an analogue of Figure 1 for the CIFAR-10 dataset. We see largely similar trends, although the magnitude of degradation is not as severe as CIFAR-100. This is consistent with our

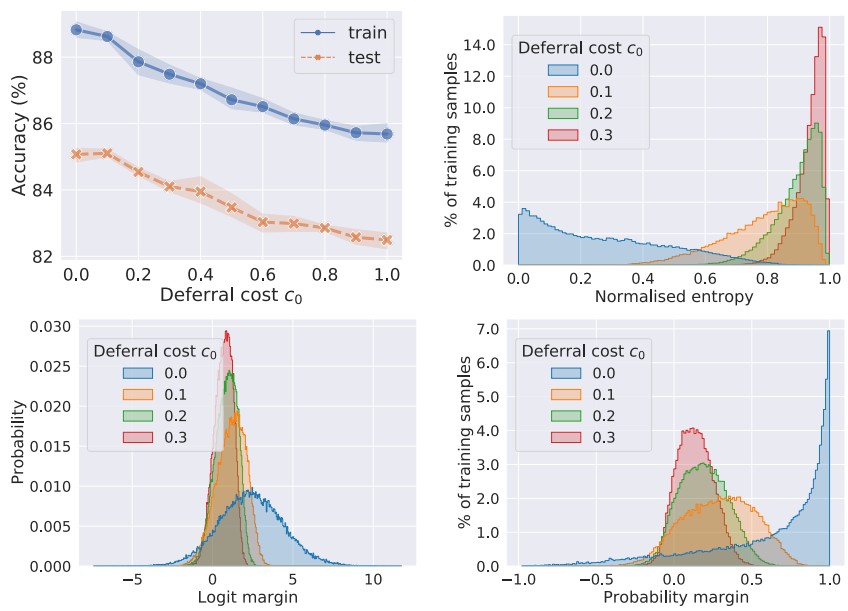

Figure 10: Illustration of underfitting of the cost-sensitive softmax cross-entropy (CSS) loss of Mozannar and Sontag [19]. We repeat the setup in Figure 1 for the **CIFAR-10** dataset. We see largely similar trends as in CIFAR-100.

analysis of the large $L$ setting being problematic, since the amount of label smoothing on all labels scales with $L$.

## J  Additional experiments: quality of one-versus-all probability estimates

We demonstrate that the probabilities $\bar{p}_1, \ldots, \bar{p}_L$ for the standard labels obtained from the one-versus-all loss (OvA) may underperform those obtained from the hybrid softmax cross-entropy plus OvA loss. Figure 11 compares the distribution of the *log-loss* $-\log \bar{p}_y(x)$, the entropy normalised by $\log L$, and the logit and probability margins. We see that the hybrid softmax cross-entropy plus OvA loss results in consistently lower log-loss values, as well as entropies and margins. Intuitively, the latter suggests that in the regular OvA loss, there are *multiple* labels $y'$ for which $\bar{f}_{y'}$ is large, resulting in multiple large $\bar{p}_{y'}$ values. As a qualifying comment, the value of OvA probabilities has been demonstrated in *out-of-domain* settings [23]; it is of interest to further study the intersection of this with the deferral setup considered in the present work.

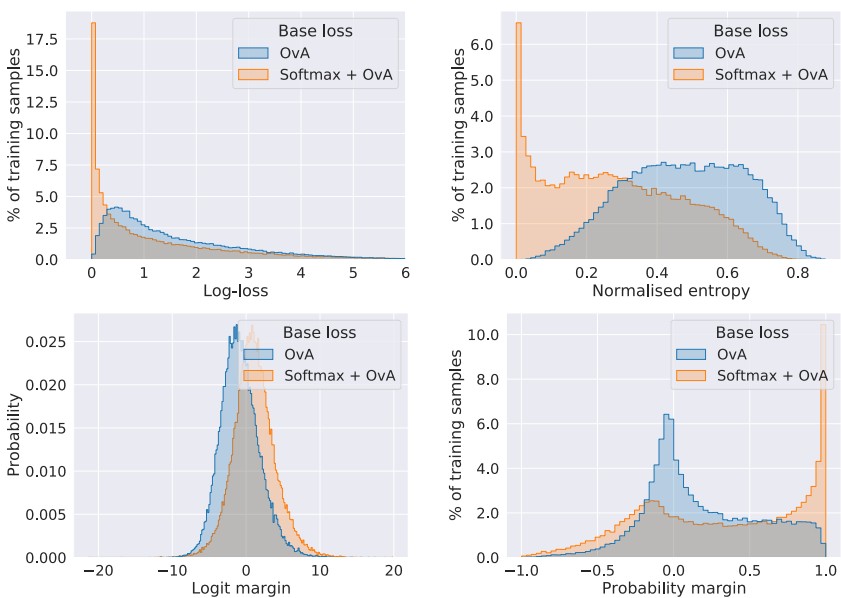

Figure 11: Comparison of standard `OvA` loss with hybrid softmax cross-entropy plus `OvA`, **CIFAR-100** dataset. We see that the hybrid softmax cross-entropy plus `OvA` loss results in consistently lower log-loss values, as well as entropies and margins.

## K Additional experiments: `OvA` and `SOvA` losses under different post-hoc schemes

We explained in §4.1 why the hybrid softmax cross-entropy plus `OvA` loss is well-suited for use with post-hoc thresholding approach, and provided empirical evidence in the previous section to showcase why it is a better alternative to the standard `OvA` loss. For the sake of completeness, we compare the performance of these losses when combined with both the post-hoc schemes proposed in this paper.

Figure 12 presents results for all four combinations (`OvA` and `SOvA` with $c_0 = 0$ coupled with either post-hoc thresholding and post-hoc training) on the CIFAR-10 and CIFAR-100 datasets in a learning to defer setting where the base models can defer to "specialist" experts (§5.2). On both datasets, it is one of the `SOvA`-based post-hoc approaches that performs the best for most operating points. Between the two post-hoc thresholding approaches, the one that uses the `SOvA` loss is clearly seen to yield significantly higher accuracies than the `OvA` loss.

We also plot the results from two additional baselines. The first is post-hoc training on a base model trained with the standard softmax cross-entropy (CE) loss. The fact that this baseline is seen to under-perform for most deferral costs highlights the benefit of using a loss that explicitly allows the base model to defer to the expert. The second is the confidence-based approach of Raghu et al. [26], where one trains a base model using the CE loss, then trains a model to get confidence estimates for the expert, and compares the base model's confidence scores with those estimated for the expert to decide whether to defer on an example. For the confidence estimation, we adopt the same architecture as the post-hoc rejector we use in our post-hoc training scheme, namely a linear model on top of the embeddings learned by the first model, and train it with the logistic loss to predict whether the expert classifies a given training example correctly. As seen, at least one of the proposed post-hoc schemes outperforms the baseline on almost all operating points, highlighting the benefit of training the defer logit jointly with the base model.

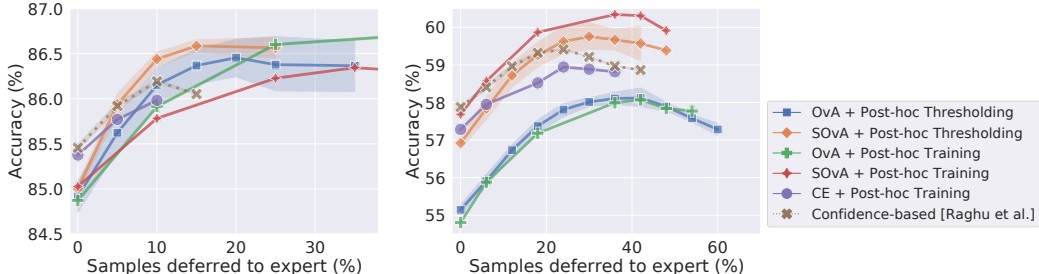

Figure 12: Additional results on CIFAR-10 (left), and CIFAR-100 (middle) in a learning to defer setting where a base model is allowed to defer to a "specialist" expert (same setting as in Figure 5). We compare the post-hoc thresholding and post-hoc training approaches with either the one-vs-all (`OvA`) loss or the hybrid softmax cross-entropy plus `OvA` loss (`SOvA`) loss used to train the base model. We additionally include post-hoc training on a base model trained with the softmax cross-entropy (CE) loss and the confidence-based approach of Raghu et al. [26]. The post-hoc training methods use the rejector parameterisations in Table 1.

## L Additional experiments: `OvA` and `CSS` losses with different post-hoc thresholding rules

In another set of experiments, we train the base model using the `OvA` and `CSS` losses (with $c_0 = 0$), and apply the post-hoc thresholding rule in (11) to defer on an example. We also include a variant of this approach, where we train using the `OvA` and `CSS` losses (with $c_0 = 0$), ignore the defer logit, and apply Chow's rule [8] with $c_{\exp}(x, y) = c_0$ to defer on an example. The latter approach does not explicitly take the expert's error into account. Figure 13 presents comparisons of all four combinations `OvA` and `CSS` losses with two the different thresholding schemes. We additionally include the standard Chow's rule for learning to reject with a constant rejection cost $c_0$ (CE + Chow), and the proposed scheme of training the base model using the `SOvA` (with $c_0 = 0$) and applying our post-hoc thresholding rule (`SOvA` + Post-hoc Thresholding). The proposed post-hoc thresholding approach is seen to yield a higher accuracy than all the baselines at most operating points. Also, we see that, as expected, the use

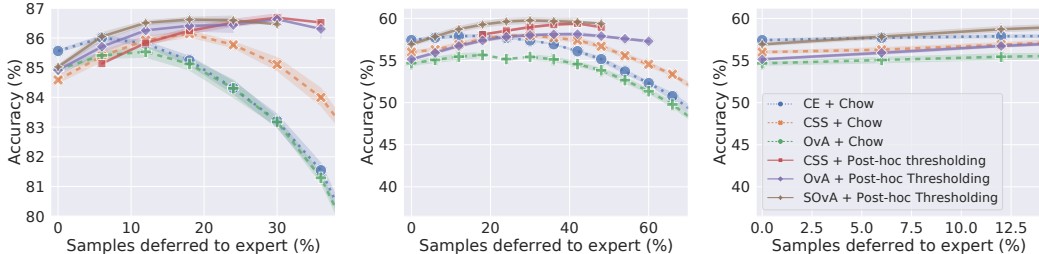

Figure 13: Additional results on CIFAR-10 (left), CIFAR-100 (middle), and ImageNet (right) in a learning to defer setting where a base model is allowed to defer to a "specialist" expert (same setting as in Figure 5). We compare for all four combinations of OvA and CSS losses (with $c_0 = 0$) coupled with either Chow's rule or the post-hoc thresholding rule in (11). We also include the standard Chow's rule where the base model is trained with the CE loss and our proposed post-hoc thresholding approach where the base model is trained with the SOvA loss.

of Chow's thresholding rule generally under-performs a post-hoc thresholding scheme that explicitly takes the expert's error into account.

## M  Additional plots: deferral risk in (1) vs $c_0$

The plots in the main body show the tradeoff between the fraction of samples deferred to the experts, and the accuracy. In these plots, the role of the fixed cost $c_0$ is *implicit*, with high fixed costs translating to a small fraction of samples deferred to the expert. We may also make the role of $c_0$ *explicit*, by studying how varying $c_0$ affects the deferral risk in (1). These are presented in Figures 14 and 15. As with Figures 5 and 6, we see that the CSS and OvA losses tend to underperform at high fixed costs $c_0$. Figures 16 and 17 further show the accuracy of the final classifier as a function of $c_0$. Again, we observe that the CSS and OvA accuracy tends to suffer as $c_0$ is increased. With the adaptive inference experiments, CSS and OvA tend to defer a large proportion of the examples to the expert even at higher costs (Figure 18), and as a result yield a higher accuracy, but with poorer deferral risk.

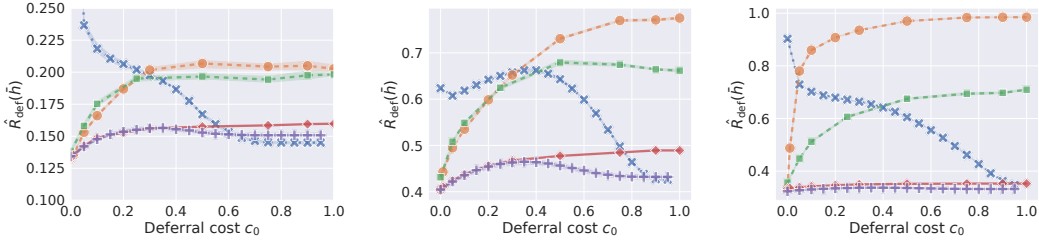

Figure 14: Results on CIFAR-10 (left), CIFAR-100 (middle), and ImageNet (right) in a learning to defer setting, where a base model is allowed to defer to a "specialist" expert. Here, we vary the fixed cost $c_0$ and study the resulting learning to defer risk (1).

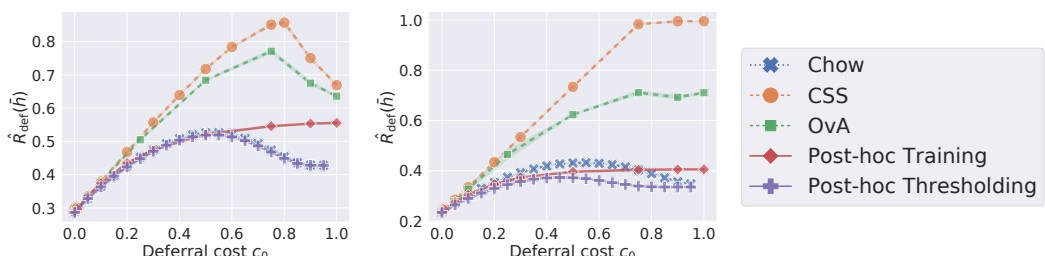

Figure 15: Results on CIFAR-100 (left), and ImageNet (right) in an adaptive inference setting, where a computationally cheap base model is allowed to defer to a more expensive expert. Here, we vary the fixed cost $c_0$ and study the resulting learning to defer risk (1).

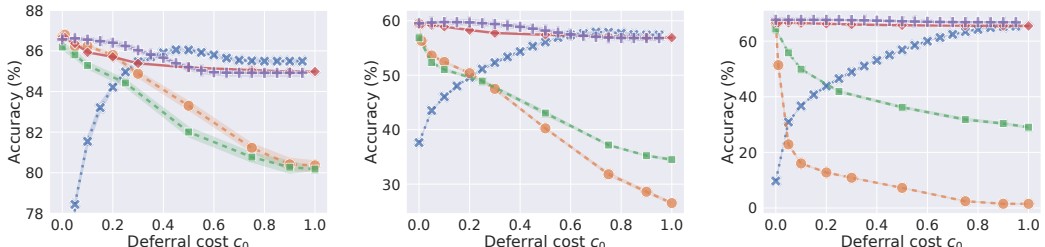

Figure 16: Results on CIFAR-10 (left), CIFAR-100 (middle), and ImageNet (right) in a learning to defer setting, where a base model is allowed to defer to a "specialist" expert. Here, we vary the fixed cost $c_0$ and study the resulting accuracy of the final classifier.

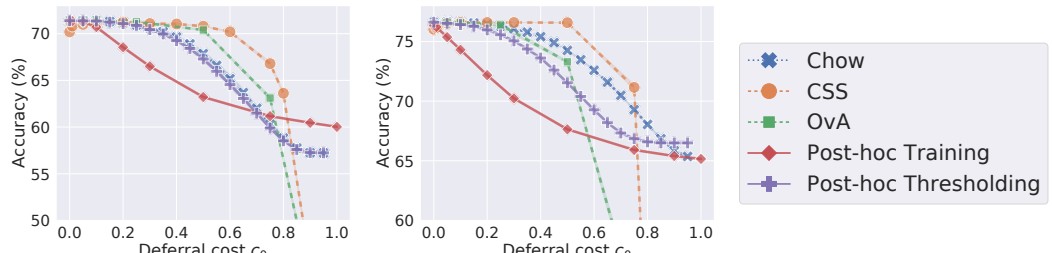

Figure 17: Results on CIFAR-100 (left), and ImageNet (right) in an adaptive inference setting, where a computationally cheap base model is allowed to defer to a more expensive expert. Here, we vary the fixed cost $c_0$ and study the resulting accuracy of the final classifier.

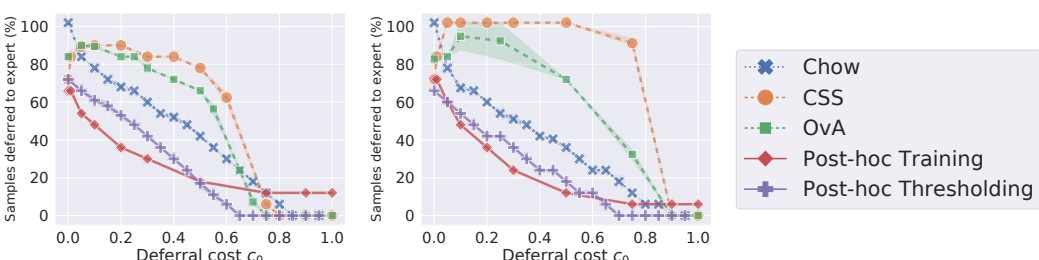

Figure 18: Results on CIFAR-100 (left), and ImageNet (right) in an adaptive inference setting, where a computationally cheap base model is allowed to defer to a more expensive expert. We vary the fixed cost $c_0$ and plot the resulting proportion of samples deferred to the expert.