# OpenReview forum: "Post-hoc estimators for learning to defer to an expert"
_NeurIPS.cc/2022/Conference — NeurIPS 2022 Accept_

### Official Review · Reviewer_sKXi · 2022-07-11

**Rating:** 7
**Confidence:** 4
**Soundness:** 3 good
**Presentation:** 3 good
**Contribution:** 2 fair

**Summary:**


The paper addresses the problem of learning to defer, in which a machine learning model learns when to defer the decision on an instance to an expert based on an estimate of the expert's cost. Existing state-of-the-art solutions are conceived for settings in which the expert cost is simply her probability of error, while real-world settings typically involve a fixed cost for querying the expert. In the latter setting, existing solutions underperform because of an oversmoothing of the prediction probabilities. The proposed solution is a couple of post-hoc strategies that prevent oversmoothing and achieve high performance in the setting involving fixed costs.

**Questions:**

My main question concerns the suboptimality of two-step learning strategies. Is it possible to say something on whether and how this affects your post-hoc estimator strategies?

**Limitations:**

There is not much discussion on the limitations of the proposed solution. My suggestion is to add something on the limitations of a two-step strategy, possibly hinting at future directions (or open problems) for a fully joint learning setting that also prevents oversmoothing.

AFTER REBUTTAL:
The authors better highlighted how the suboptimality of the two step procedure is (for threshold correction) or could be (for rejector training) quantified and possibly addressed.


**Strengths And Weaknesses:**

PROs:

The paper identifies a structural limitation of recent solutions that seriously affects their applicability to real-world scenarios.

The proposed solution is theoretically grounded and easy to implement.

A careful experimental evaluation confirms the substantial advantage of the proposed post-hoc approaches wrt sota solutions.

CONs:

The solution is a post-hoc strategy, in which the base model (and the
expert cost estimator, in the thresholding case) is trained in a
simpler setting (zero fixed cost), and the aggregate model is adjusted
post-hoc on the desired fixed cost c0. This two-step strategy is
usually suboptimal, as the model is not optimized for its working
conditions.

---

> ### Author Response · Authors · 2022-08-02
> **Response to Reviewer sKXi**
>
> Thanks for the detailed feedback. We have uploaded a revision that incorporates your suggestions.
>
> > *My main question concerns the suboptimality of two-step learning strategies. Is it possible to say something on whether and how this affects your post-hoc estimator strategies?*
>
> This is a good question. Note that for threshold correction (Section 4.1), we **do have precisely such a guarantee**: in Proposition 2, we consider the effect of using a potentially sub-optimal classifier and/or estimator of the expert’s error rate. Intuitively, the discrepancy of these quantities to the true values (in an L1 sense) control the error rate of the final model.
>
> For rejector training (Section 4.2), we agree that a similar bound would be useful. Note that this is somewhat complicated by the presence of two learned quantities, the rejector and classifier. In **Appendix A.1** (Lemma 4), we have sketched how one can get some handle on the excess risk of this approach. However, it does not capture the potential sub-optimality of the rejector. Studying this closer could be an interesting avenue for future work.
>
>
> > *There is not much discussion on the limitations of the proposed solution. My suggestion is to add something on the limitations of a two-step strategy, possibly hinting at future directions (or open problems) for a fully joint learning setting that also prevents oversmoothing.*
>
> Thanks for the suggestion. We have expanded the discussion of potential limitations of our strategy, and offered suggestions on potential directions for future work (**Section 6**). In particular, we have highlighted the potential use of a joint surrogate loss based on (Lee et al., ‘04) to prevent over-smoothing.

---

> > ### Comment · Reviewer_sKXi · 2022-08-08
> > **Response to authors**
> >
> > Thanks for the clarifications and the additions to the manuscript. I think they contribute to strengthen the paper, and I raised my score accordingly.

---

### Official Review · Reviewer_xmUy · 2022-07-12

**Rating:** 6
**Confidence:** 4
**Soundness:** 2 fair
**Presentation:** 3 good
**Contribution:** 2 fair

**Summary:**

This paper tackles the problem of jointly learning a classifier and a rejector where the rejector can choose whether the classifier or the human should predict, this problem is denoted as learning to defer and the authors tackle the problem when the cost of deferring to the expert is a constant $c_0$ plus the misclassification error of the expert. The authors show that previous state-of-the-art approaches fail when $c_0>0$ by making a connection to label smoothing. They then propose two training strategies for the rejector that combine previous surrogate approaches with new machinery. They evaluate their method on synthetic and semi-synthetic data (CIFAR-10 with simulated experts) and show that they outperform previous methods.

**Questions:**

Why are Figure 5 and Figure 6 showing performance on accuracy instead of on the objective in equation 1?  The baselines optimize equation 1 instead of human-AI accuracy, and it is not fair to show only results of accuracy, but one must show results on the loss in equation 1. Furthermore, you must show that the approach does better on the loss in equation 1 and on accuracy.

What is the performance on Figure 5 of cSS and OvA when you set $c_0=0$ and you just vary the coverage by changing the deferral threshold ?

**Limitations:**

The authors address the limitations adequately.

**Strengths And Weaknesses:**

Originality: The paper has novel analysis of previous surrogate approaches for learning to defer (L2D) when the cost of deferral has an additional penalty $c_0$. The paper proposes a new method to handle this case which leverages previous methods, namely OvA, and adds an extra term in the loss function to form it. This new surrogate has limited novelty and doesn't address the case where $c_0$ is potentially a function of X, a more realistic extension. The related work is very well cited.

Quality: I checked the proofs for the theoretical results and the claims in the papers and they are sound. The empirical results seem reasonable, but no code was provided to verify the results. Furthermore, there are more baselines in the literature that should have been performed. Most notably, the Confidence baseline, where one trains 1) a classifier using cross entropy, 2) a model for expert error, and then 3) defers if classifier error is higher than expert error + $c_0$. One important concern, is that the baselines of Mozannar and Sontag and that of OvA optimize for the cost sensitive loss, while the results show in the experiments  accuracy (without the penalty c_0).  Furthermore, the paper only has experiments with synthetic human experts which is not sufficient to back up claims for human-AI performance.

Clarity: Please make section titles and the title have each word start with a uppercase letter. The paper is well written and easy to follow.

Significance: I think the paper makes an important contribution in showing that previous surrogate approaches may underfit when $c_0>0$. However, one concern is that the paper doesn't show that the method optimizes for the objective in equation 1, but only that the baselines underfit the target. I think if the authors can answer the questions below in the positive, then my overall evaluation may be raised substantially.

---

> ### Author Response · Authors · 2022-08-02
> **Response to Reviewer xmUy**
>
> Thanks for the detailed feedback. We have uploaded a revision that includes the additional experimental results requested.
>
> > *One important concern, is that the baselines of Mozannar and Sontag and that of OvA optimize for the cost sensitive loss, while the results show in the experiments accuracy (without the penalty c_0)… However, one concern is that the paper doesn't show that the method optimizes for the objective in equation 1, but only that the baselines underfit the target… Why are Figure 5 and Figure 6 showing performance on accuracy instead of on the objective in equation 1? The baselines optimize equation 1 instead of human-AI accuracy, and it is not fair to show only results of accuracy, but one must show results on the loss in equation 1. Furthermore, you must show that the approach does better on the loss in equation 1 and on accuracy.*
>
> We wish to clarify that Figures 5 and 6 plot the **deferral-accuracy tradeoff**: the x-axis is the fraction of samples deferred to the expert, and the y-axis is the accuracy. Importantly, the x-axis is **not the same as c0**: instead, we vary c0 for each method, and compute the fraction of samples deferred. Such a performance summary has previously been considered (e.g., Figure 5, 9 in Mozannar and Sontag).
>
>
> Nonetheless, we are happy to also include plots of the fixed deferral cost c0 versus the overall deferral risk (Equation 1), viz. the accuracy plus the overall deferral cost. These have been added in **Appendix N** (Figures 15 and 16). These show the same general trends as the accuracy-deferral plots: CSS and OvA tend to perform poorly (i.e., yield higher deferral risks) under high fixed costs c0.
>
>
> > *What is the performance on Figure 5 of cSS and OvA when you set c0=0 and you just vary the coverage by changing the deferral threshold ?*
>
>
> Please note that **Appendix K** (Figure 12) showed the performance of OvA + post-hoc thresholding. For CSS + post-hoc thresholding, please see **Appendix L** (Figure 13). Note that SOvA + post-hoc thresholding generally outperforms both approaches, as per the discussion in Line 186 - 189.
>
> > *Furthermore, there are more baselines in the literature that should have been performed. Most notably, the Confidence baseline, where one trains 1) a classifier using cross entropy, 2) a model for expert error, and then 3) defers if classifier error is higher than expert error +
> C0.*
>
> Thanks for the suggestion. Please note that in **Appendix K** (Figure 12), we reported results from a method that trains a base model with cross-entropy, and then trains a post-hoc rejector. This can be seen as combining Steps 2 – 3 into a single optimisation problem.
>
> We have now also added exactly the baseline suggested by the reviewer in **Appendix M** (Figure 14). As seen, the proposed post-hoc schemes outperforms this baseline at most operating points.
>
>
> > *Furthermore, the paper only has experiments with synthetic human experts which is not sufficient to back up claims for human-AI performance*
>
> Please note that the L2D paradigm was intended to be just one application of our framework. The adaptive inference paradigm, wherein the experts are larger ML models, is another important motivation for our work. We evaluated methods under this paradigm in Figure 6. More broadly, there may be scenarios where the expert is a ML model trained on a slightly different distribution from the base model (which is similar to our “specialist” expert setup in Figure 5). In such cases, it's important that we selectively forward examples to the expert.
>
> > *This new surrogate has limited novelty and doesn't address the case where  c0 is potentially a function of X, a more realistic extension.*
>
> Please note that **Appendix D** (Equation 23 in revision; first equation in original submission) in fact provides an extension for the case of instance-dependent cost. We made a note in passing about this in the body (L148), but chose to focus on the instance-independent case since this has been the primary setting of prior work.
>
> > *Please make section titles and the title have each word start with a uppercase letter.*
>
> This has been fixed.

---

> > ### Author Response · Authors · 2022-08-08
> > **Re: Response to Reviewer xmUy**
> >
> > Thanks again for the detailed feedback. As the discussion period is ending shortly, we wanted to check if you had a chance to look at our response (+ revised submission). We have tried to address the comments from your initial review, but would be happy to discuss any points further.

---

### Official Review · Reviewer_3cKz · 2022-07-13

**Rating:** 7
**Confidence:** 3
**Soundness:** 4 excellent
**Presentation:** 4 excellent
**Contribution:** 2 fair

**Summary:**

  This paper augments the CSS and OvA loss functions used for learning to defer such that they
  no longer underfit the base model, particularly in problems with many possible labels. This is
  done through the introduction of two post-hoc estimators which can used to take a calibrated
  model where there is zero cost in consulting an expert and adapt it for the setting where
  there is a positive cost for deferring to an expert. There are both theoretical and experimental
  results to support the contributions.


**Questions:**

What rejector losses should be preferred for their settings?

**Limitations:**

The authors do not adequately address the limitations and societal impact of the work. While I don't think there is significant ethical or negative societal risk with this work, the paper would have been strengthened by some discussion of the limitations of the presented methods.

**Strengths And Weaknesses:**

  Originality:

  The particular algorithms introduced in the paper appear to be novel. Post-hoc procedures such
  as this one do occur for uncertainty calibration, but this is a novel application of that approach.


  Technical Quality:

  The technical quality is very good with the experiments well-thought out and providing very compelling
  results. Since this work is meant to used on top of CSS or OvA it really needed to be compared with
  those methods for accuracy or loss. The proofs for calibration of the loss function were also clear.
  Some of the results are left a little open-ended. When calibrating with a rejector loss there are many
  to choose from and there is not much insight as to why one loss should be preferred over another.

  Clarity:

  The entire presentation of this paper is written very clearly and easy to follow. Both in the main presentation
  and in the appendix it was easy to understand what was being done in each experiments and the proofs
  was easy to follow as well.

  Significance:

  This work is very similar to the loss of Verma and Nalisnick and while it is presented as a generalisation
  of that work it's not immediately clear that it's a significant contribution by itself. I nevertheless suspect
  the post-hoc methods introduced in the paper would be useful for others to build on top of.

---

> ### Author Response · Authors · 2022-08-02
> **Response to Reviewer 3cKz**
>
> Thanks for the detailed feedback. We are heartened by the comment that “the technical quality is very good with the experiments well-thought out and providing very compelling results”.
>
> > *This work is very similar to the loss of Verma and Nalisnick and while it is presented as a generalisation of that work it's not immediately clear that it's a significant contribution by itself.*
>
> While our work certainly builds on that of Verma and Nalisnick (amongst others), there are a few differences that we believe are non-trivial:
> We identify and explain the underfitting phenomenon of the CSS loss in (Mozannar and Sontag, ‘20), as well as the OvA loss in (Verma and Nalisnick, ‘22), when there is non-zero fixed deferral cost c0 (Section 3.1, 3.2). **This issue has not been identified in any prior work** (to our best knowledge), and **we believe is non-trivial** (as it identifies a failing of these losses in a natural scenario).
>
>
> We propose post-hoc estimators for the case of non-zero fixed deferral cost (Section 4), and analyze their theoretical behavior (Proposition 2, Lemma 3). These estimators are considerably different from the end-to-end OvA loss proposed in (Verma and Nalisnick, ‘22), while being practically performant in the case of c0 > 0.
>
>
> In the course of our analysis, we also extend the OvA loss in Verma and Nalisnick to handle the case of non-zero fixed deferral cost (Equation 10), and establish the consistency of the same (Lemma 1).
>
> > *When calibrating with a rejector loss there are many to choose from and there is not much insight as to why one loss should be preferred over another… What rejector losses should be preferred for their settings?*
>
> For the post-hoc rejector training in Section 4.2, in principle any classification-calibrated loss function for binary classification is admissible. As with standard binary classification tasks, the hinge and logistic losses are reasonable defaults, and we indeed found minimal benefit in altering this loss in our experiments. Further constraints (e.g., label noise, label imbalance) would of course motivate alternate losses.
>
> > *Since this work is meant to used on top of CSS or OvA it really needed to be compared with those methods for accuracy or loss.*
>
> To clarify, in Figures 5 and 6, we have compared against both CSS and OvA in terms of the tradeoff between accuracy and fraction of samples deferred.

---

> > ### Comment · Reviewer_3cKz · 2022-08-08
> > **Thanks for the response**
> >
> > I wish to thank the authors for their response and clarifications. I have read over the updated paper and supplemental material. The paper is much improved and I remain confident in my assessment.

---

### Author Response · Authors · 2022-08-02
**Common response to all reviewers**

Thanks all for the detailed comments! We have uploaded a revision based on the reviewers’ suggestions, namely:
- Added plots (**Appendix N**) that show the behaviour of the deferral risk (Equation 1) as a function of c0. These complement the existing tradeoff plots in Figure 5 and 6.
- Added results for Confidence baseline (**Appendix M**).
- Expanded results for using CSS/OvA loss with different post-hoc thresholding schemes (**Appendix L**). This expands the original results in Appendix K to include the CSS loss.
- Added excess risk bound for post-hoc training (**Appendix A.1**). This supplements the existing excess risk bound for post-hoc thresholding.
- Expanded discussion of limitations and future work (**Section 6**).
- Updated casing of section titles to capitalise first letter of each word.

Barring the expanded text in Section 6, these changes are all in the appendices.

---

### Meta-Review · Area_Chair_nxaw · 2022-08-27

**Recommendation:** Accept
**Confidence:** Certain

**Metareview:**

This paper proposed a novel framework for “learning to defer” (L2D), which decides when to defer the decision on an instance to an expert based on an estimate of the expert's cost. The key results include identifying the “failure mode” of existing L2D approaches, a novel post-doc estimation procedure for model calibration, and thorough experiments showing that the proposed algorithm works well when compared to SOTA. During the rebuttal phase, the authors included additional experiments which make the empirical performance more convincing (e.g., baselines + post-hoc thresholding, and additional baseline results as suggested by Reviewer xmUy). Other than several clarity issues, there were no critical concerns in the reviews.

There are valuable suggestions in the reviews, including improving the clarity when introducing key concepts and notations in the main text, and providing details of the experimental setting and results. The authors are strongly encouraged to address the concerns raised in the reviews when preparing a revision of this paper.


**Award:**

No

---

### Decision · Program_Chairs · 2022-09-14

Accept